


Combining livestock production information in a process based vegetation model to reconstruct the history of grassland management

Jinfeng Chang[1,2], Philippe Ciais[1], Mario Herrero[3], Petr Havlik[4], Matteo Campioli[5], Xianzhou Zhang[6], Yongfei Bai[7], Nicolas Viovy[1], Joanna Joiner[8], Xuhui Wang[9,10], Shushi Peng[10], Chao Yue[1,11], Shilong Piao[10], Tao Wang[11], Didier A. Hauglustaine[1], Jean-Francois Soussana[12]

[1]Laboratoire des Sciences du Climat et de l'Environnement, UMR8212, CEA-CNRS-UVSQ, 91191 Gif-sur-Yvette, France

[2]Sorbonne Universités (UPMC, Univ Paris 06)-CNRS-IRD-MNHN, LOCEAN/IPSL, 4 place Jussieu, 75005 Paris, France

[3]Commonwealth Scientific and Industrial Research Organisation, Agriculture Flagship, St. Lucia, QLD 4067, Australia

[4]Ecosystems Services and Management Program, International Institute for Applied Systems Analysis, 2361 Laxenburg, Austria

[5]Centre of Excellence PLECO (Plant and Vegetation Ecology), Department of Biology, University of Antwerp, 2610 Wilrijk, Belgium

[6]Lhasa Plateau Ecosystem Research Station, Key Laboratory of Ecosystem Network Observation and Modeling, Institute of Geographic Sciences and Natural Resources Research, CAS, Beijing 100101, China

[7]State Key Laboratory of Vegetation and Environmental Change, Institute of Botany, Chinese Academy of Sciences, Beijing 100093, China

[8]NASA Goddard Space Flight Center, Greenbelt, MD, USA

[9]Laboratoire de Météorologie Dynamique, Institute Pierre Simon Laplace, 75005 Paris France

[10]Sino-French Institute of Earth System Sciences, College of Urban and Environmental Sciences, Peking University, 100871 Beijing China

[11]CNRS and UJF Grenoble 1, UMR5183, Laboratoire de Glaciologie et Géophysique de l'Environnement (LGGE), Grenoble, France

[12]INRA, UAR 0233 CODIR Collège de Direction. Centre-Siège de l'INRA, Paris , France.

Correspondence to: J. Chang (jinfeng.chang@locean-ipsl.upmc.fr)





**Abstract.** Grassland management type (grazed or mown) and intensity (intensive or extensive) play a crucial role in the GHG balance and surface energy budget of this biome, both at field scale and at large spatial scale. Yet, global gridded historical information on grassland management intensity is not available. Combining modelled grass biomass productivity with statistics of the grass-biomass demand by livestock, we reconstruct gridded maps of grassland management intensity from 1901 to 2012. These maps include the minimum area of managed vs. maximum area of un-managed grasslands, and the fraction of mown versus grazed area at a resolution of 0.5° by 0.5°. The grass-biomass demand is derived from a livestock dataset for 2000, extended to cover the period 1901 – 2012. The nature of grass-biomass supply (i.e., forage grass from mown grassland and biomass grazed) is simulated by the process based model ORCHIDEE-GM driven by historical climate change, rising $CO_2$ concentration, and changes in nitrogen fertilization. The global area of managed grassland obtained in this study is simulated to increase from $5.1 \times 10^6$ km$^2$ in 1901 to $11 \times 10^6$ km$^2$ in 2000, although the expansion pathway varies between different regions. The gridded grassland management intensity maps are model-dependent because they depend on Net Primary Productivity (NPP), which is the reason why specific attention is given to the evaluation of NPP. Namely, ORCHIDEE-GM is calibrated for C3 and C4 grass functional traits, and then evaluated against a series of observations from site-level NPP measurements to two global satellite products of Gross Primary Productivity (GPP) (MODIS-GPP and SIF data). The distribution of GPP and NPP with and without management, are evaluated against observations at different spatial and temporal scales. Generally, ORCHIDEE-GM captures the spatial pattern, seasonal cycle and interannual variability of grassland productivity at global scale well, and thus appears to be appropriate for global applications.

## 1 Introduction

The rising concentrations of greenhouse gases (GHGs), such as carbon dioxide (CO2), methane (CH4), and nitrous oxide (N2O) are driving climate change, through increased radiative forcing (IPCC, 2013). It is estimated that over the whole globe, livestock production (including crop-based and pasture-based) currently accounts for 37% of the anthropogenic CH$_4$ (Martin et al., 2010) and 65% of the anthropogenic N$_2$O emissions (FAO, 2006). Grassland ecosystems support most of the world's livestock production, thus contributing indirectly a significant share of global CH$_4$ and N$_2$O emissions. For CO$_2$ fluxes however, grassland can be either a sink or a source with respect to the atmosphere. The net annual carbon storage of managed grassland ecosystems in Europe (here, net biome productivity, NBP) was found to be correlated with carbon removed by grazing and/or mowing (Soussana et al., 2007). Thus, knowledge of management type (grazed or mown) and intensity (intensive or extensive) is crucial for simulating the carbon stocks and GHG fluxes of grasslands.

The HYDE 3.1 land-use dataset (Klein Goldewijk et al., 2011) provides reconstructed gridded changes of *pasture* area over the past 12,000 years. Here, *pasture* represents managed grassland providing grass biomass to livestock. This reconstruction is based on population density data and country-level per





capita use of pasture land derived from FAO statistics (FAO, 2008) for the post-1961 period or assumed by those authors for the pre-1960 period. It defined land used as pasture but does not provide information about management intensity. To our knowledge, global maps of grassland management intensity are not available. Chang et al. (2015a) constructed such a map for European grasslands at 25

km spatial resolution based on livestock numbers from statistics, and the grass-fed livestock numbers supported by the net primary productivity (NPP) of the ORCHIDEE-GM model. The main result of that study is that soil carbon accumulation is accelerating in European grassland, with a net increase of soil carbon of $384 \pm 141$ g C m$^{-2}$ over the period 1991- 2010 (Chang et al., 2015a). The increasing soil carbon accumulation rate was attributed separately to climate change, $CO_2$ trends, nitrogen addition,

and land-cover and management intensity changes. The observation-driven trends of management intensity were found to be the dominant driver explaining the positive trend of NBP across Europe (36 - 43% of the total trend with all drivers; Chang et al., 2015c). That study confirmed the importance of management intensity in drawing up a grassland carbon balance. Despite being carbon sinks, the European grassland was found to be a net GHG source of 50 g C-$CO_2$ equiv. m$^{-2}$yr$^{-1}$ because $CH_4$ and

$N_2O$ emissions, and $CO_2$ released by animals (Chang et al., 2015a) offset soil carbon accumulation. This study illustrated the importance of accounting for not only the ecosystem GHG fluxes, but also the livestock-related fluxes, when estimating the GHG balance of grassland.

Recently, Herrero et al. (2013) garnered a global livestock data to create a dataset with grass biomass

use information for year 2000. In this dataset, grass used for grazing or silage is separated from grain feeds, occasional feeds and stovers (fibrous crop residues). A variety of constraints have been taken into account in creating this global dataset, including the specific metabolisable energy requirements for each animal species, and regional differences in animal diet composition, feed quality and feed availability.  This grass-biomass use dataset provides a starting point for constraining the amount of

carbon removed by grazing and mowing, and is suitable for adoption by global vegetation models to account for livestock-related fluxes.

The major objective of this study is to produce global gridded maps of grassland management intensity since 1901 for global vegetation model applications. These maps combine historical NPP changes from

the process-based global vegetation model ORCHIDEE-GM (Chang et al., 2013; 2015b) with grass biomass use extrapolated from Herrero et al. (2013). First, ORCHIDEE-GM is calibrated to simulate the distribution of *potential* (maximal) harvested biomass from mown and grazed grasslands. Second, the calibrated model is evaluated against both a new set of site-level NPP measurements, and satellite-based models of NPP and GPP. In a third step, the modelled NPP maps are used in combination with

livestock data in each country since 1961 and in 18 large regions of the globe for 1901-1960 for reconstructing annual maps of grassland management intensity at a spatial resolution of 0.5° by 0.5°. The reconstructed management intensity defines the fraction of mown, grazed and unmanaged grasslands in each grid-cell. In Sect. 2, we describe the ORCHIDEE-GM model, the adjustment of its parameters for the C4 grassland biome, the data used for evaluation, and the method proposed to

reconstruct grassland management intensity. The management intensity maps and the comparison





between modelled and observed productivity are presented in Sect. 3 and discussed in Sect. 4. Concluding remarks are made in Sect. 5.

## 2 Material and Methods

### 2.1 Model description

ORCHIDEE (ORganizing Carbon and Hydrology In Dynamic Ecosystems) is a process-based ecosystem model developed for simulating carbon fluxes, and water and energy fluxes in ecosystems,

from site-level to global scale (Krinner et al., 2005; Ciais et al., 2005; Piao et al., 2007). ORCHIDEE-GM (Chang et al., 2013) is a version of ORCHIDEE that includes the grassland management module from PaSim (Reido et al., 1998; Vuichard et al., 2007a,b; Graux et al., 2011), a grassland model for field-scale applications. ORCHIDEE-GM v1 was evaluated and some of its parameters calibrated, at 11 European grassland sites representative of a range of management practices, with eddy-covariance net

ecosystem exchange (NEE) and biomass measurements. The model successfully simulated the NBP of these managed grasslands (Chang et al., 2013). At continental scale, ORCHIDEE-GM version 2.1 was applied over Europe to calculate the spatial pattern, interannual variability (IAV) and the trends of potential productivity, i.e., the productivity of an optimal management system that maximizes simulated livestock densities in each grid-cell (Chang et al., 2015b). Chang et al. (2015b) then added a

parameterization of adaptive management through which farmers react to a climate-driven change of previous-year productivity. Though a full nitrogen cycle is not included in ORCHIDEE-GM, the positive effect of nitrogen fertilizers on grass photosynthesis rates, and thus on subsequent ecosystem productivity and carbon storage, are parameterized with an empirical function calibrated from literature estimates (Chang et al., 2015b). ORCHIDEE-GM v2.1 was used to simulate NBP and NBP trends over

European grasslands during the last five decades at a spatial resolution of 25 km and a 30-minute time-step.

ORCHIDEE-GM v1 and v2.1 were developed based on ORCHIDEE v1.9.6. To benefit from recent developments and bug-corrections in the ORCHIDEE model, ORCHIDEE-GM v2.1 is updated in this

study with ORCHIDEE Trunk.rev2425 (available at: https://forge.ipsl.jussieu.fr/orchidee/browser/trunk#ORCHIDEE). The updated model is referred to here as ORCHIDEE-GM v3.1.

### 2.2 Model parameter settings

Two sensitive parameters representing photosynthetic activity (the maximum rate of Rubisco carboxylase activity at a reference temperature of 25°C; $Vc_{max}25$) and the morphological plant traits (the maximum specific leaf area; $SLA_{max}$) were reported by Chang et al. (2015a) for simulating grassland NPP. The $Vc_{max}25 = 55$ μmol m$^{-2}$ s$^{-1}$ and $SLA_{max} = 0.048$ m$^2$ per g C in ORCHIDEE-GM were

previously defined from observations and indirectly evaluated against eddy-flux tower measurements





of GPP for temperate C3 grasslands in Europe (Chang et al., 2013, 2015b). For C4 grasslands, we set the value of $SLA_{max}$ = 0.044 m$^2$ per g C for C4 grasses in ORCHIDEE-GM to fit the mean value from the TRY TRY global plant trait database (0.0403 m$^2$ per g C) as we did previously for C3 grasses (Chang et al., 2013). $Vc_{max}25$ for C4 grasses is set to be 25 μmol m$^{-2}$ s$^{-1}$ (Feng and Dietze, 2013; Verheijen et al., 2013). These values of $Vc_{max}25$ represent an average for different nitrogen, phosphorus conditions between locations, and for different species of C4 grasses possibly adapted to specific long-term climate conditions at each location, They are within the range of observations made under different conditions, and consistent with values used by other terrestrial ecosystem models (Table S1).

### 2.3 Simulation set-up

ORCHIDEE-GM v3.1 was run on a global grid over the globe using the CRU+NCEP reconstructed climate data for the period 1901–2012 (http://dods.extra.cea.fr/data/p529viov/cruncep/readme.htm). The fields used as input of the model are temperature, precipitation, specific humidity, solar radiation, wind speed, pressure and long wave radiation at a 6-hourly time-step. The CRU+NCEP climate is a combination of CRU TS.3.21 0.5° × 0.5° monthly climate fields covering the period 1901–2012 (http://badc.nerc.ac.uk/view/badc.nerc.ac.uk__ATOM__dataent_1256223773328276), and the US National Oceanic and Atmospheric Administration (NOAA) National Centers for Environmental Prediction (NCEP) and National Center for Atmospheric Research (NCAR) reanalysis 1° × 1° 6-hourly climatology covering the period 1948 to the present-day (Kanamitsu et al., 2002).

Other input data are: 1) yearly grazing-ruminant stocking density maps, 2) wild-herbivores population density maps, 3) nitrogen (N) fertilizer application maps including manure-N and mineral-N fertilizers, and 4) atmospheric N deposition maps. These input maps all cover the period from 1901 to 2012 and are briefly described below (see Supplementary Information Text S2 – S8).

**Grazing livestock density maps**. These maps are established from animal density, and grassland land-use area data (see Supplementary Information Text S1, S2, and S3). The ruminant livestock density distribution for year 2006 is from the Gridded Livestock of the World v2.0 dataset (GLW v2.0). Domestic ruminant stocking densities including cattle, sheep and goats (Robinson et al., 2014) are converted to Livestock Units (LU) and aggregated to the resolution of 0.5° × 0.5°. This gridded ruminant density is then back-casted from 2012 to 1901 assuming that it has changed in each grid-cell proportionally with country-scale metabolisable energy requirement (ME) from all ruminants (Supplementary Information Text S2 and S3). ME requirement is the amount of energy (MJ day$^{-1}$) an animal needs for maintenance, lactation, and pregnancy (IPCC, 2006 Vol 4, Chapter 10, Eqs. 10.3 to 10.13). The evolution of ME requirement by ruminants was calculated from FAO ruminant population statistics during the period 1961-2012 (FAOSTAT, 2013) and from Mitchell (1993, 1998a, b) during the period 1901-1960 (http://themasites.pbl.nl/tridion/en/themasites/hyde/landusedata/livestock/index-2.html) using the method given in the Supporting information Text S1 of Chang et al. (2015b).





**Wild herbivore density maps.** The gridded population of wild herbivores is derived from the literature data, and from Bouwman et al. (1997) (see Table S2). The population of these herbivores was first converted to LU according to the ME requirement calculated from their mean weight (Table S2), and then distributed to non-managed grasslands based on grassland aboveground (consumable) NPP

simulated from ORCHIDEE-GM v3.1 (Supplementary Information Text S8). The wild herbivores density was assumed to remain constant during the period of 1901-2012, because no gridded worldwide wild-animals population information was available.

**Nitrogen application rates from mineral fertilizers and manure.** For grasslands in the EU-27

countries, gridded mineral fertilizer and manure nitrogen application rates for grasslands are available from the CAPRI model (Leip et al., 2011, 2014) based on information from official and harmonized data sources such as Eurostat, FAOstat and OECD, which are spatially disaggregated using the methodology described by Leip et al. (2008). The rules used to rebuild the temporal evolution of gridded nitrogen fertilization from 1901 to 2010 were described by Chang et al., 2015b, namely: 1) no

mineral-N fertilizer is applied over grasslands before 1950, and 2) for the period of 1951-1961, the rate of application is assumed to increase linearly from zero to the level of 1961 in each 0.5° grid-cell. The application rate in 2011 and 2012 was assumed to be constant and the same as that in 2010. For countries/region outside the EU-27, the following data and methods were used (see Supplementary Information Text S4 and S5 for details). The regional amount of manure-N fertilizer from Bouwman et

al. (2002a, b) was downscaled to a 0.5° × 0.5° grid according to ruminant density of each grid-cell, which implies that locally higher ruminant density produces more manure. In each grid-cell, historical changes of manure-N fertilization were assumed to follow the same evolution as the ruminant density (Supplementary Information Text S2).

For mineral-N fertilizers, country-scale average application rates and the grassland areas where mineral-N fertilizers have been applied from 1999 to 2000 are taken from FAO/IFA (2002). National application rates are downscaled at 0.5° × 0.5° resolution assuming that only grid-cells with a ruminant density above a certain threshold are fertilized with mineral fertilizers. The value of this threshold is determined for each country so that the total grassland area of fertilized grids is identical to the national

fertilized grassland area reported by FAO/IFA (2002). The application rate of mineral-N fertilizers is extrapolated using country-scale total nitrogenous mineral fertilizers consumption data (TNF) from FAOSTAT (2013) during the period 1961-2002. The mineral-N fertilization rate after 2002 is assumed to be constant as the 2002 rate. For the period 1901-1960, the same set of rules that were applied for the EU-27 (see section 'Simulation set-up' in Chang et al., 2015a for details) is used.

**Atmospheric-nitrogen deposition maps.** The historical atmospheric N deposition maps were simulated by the LMDz-INCA-ORCHIDEE global chemistry-aerosol-climate model which couples on-line the LMDz (Laboratoire de Météorologie Dynamique, version-4) General Circulation Model, the INCA (INteraction with Chemistry and Aerosols, version-3) chemistry transport model and

ORCHIDEE v9 dynamical vegetation model. A description of the model components is given by





Hauglustaine et al. (2014). Hindcast simulations for the years 1850, 1960, 1970, 1980, 1990, and 2000, have been performed using anthropogenic emissions from Lamarque et al. (2010). Based on these simulations, the LMDz-INCA total nitrogen deposition fields (wet and dry ; NHx and NOy) of all nitrogen-containing gas phase and aerosol species have been simulated at a spatial resolution of $1.9^o$ in

latitude and $3.75^o$ in longitude. These deposition fields have been evaluated against measurements from the EMEP network over Europe (emep.int), from the NADP network over North America (http: //nadp.sws.uiuc.edu/NTN) and from the EANET network over eastern Asia (http://www.eanet.cc/). They show a generally good agreement with observations (Hauglustaine et al., 2014). Linear interpolation was performed between the hindcasts years to produce temporally variable atmospheric-N

deposition maps.

In this study, we first model the productivity of grasslands at global scale by ORCHIDEE-GM v3.1, and evaluate that its distribution is realistic using local and satellite observations. Then, we derive historical maps of management intensity from productivity maps. Thus, we do not use a land-cover

map in the simulations, but rather consider that grasslands are distributed all over the world. Considering different photosynthetic pathways and management types, six grassland plant functional types (PFTs) are separately defined:  C3 natural (unmanaged) grassland, C3 mown grassland, C3 grazed grassland, C4 natural (unmanaged) grassland, C4 mown grassland, and C4 grazed grassland. ORCHIDEE-GM v3.1 is run over the globe during the period 1901-2012 with those six PFTs being

present in each grid-cell, forced by increasing $CO_2$, variable climate and variable nitrogen deposition. For each grassland PFT, specific forcing and management strategies are used (summarized in Table 1). Unmanaged grasslands are forced by wild herbivore density maps and grazing rates, which consider both green biomass grazing in the growing season and dead biomass grazing in the non-growing season (Supplementary Information Text S8 for detail). Both mown and grazed grassland are forced by the

historical N fertilizer maps described above, which include manure and mineral fertilizers. In the mown grassland, the frequency and magnitude of regular harvests of forage in each grid-cell during the growing season is simulated internally in the model as a function of grown biomass (Vuichard et al., 2007). The annual production of forage from the mown grassland fraction of a grid-cell is defined as the *potential* biomass that can be cut (Chang et al., 2015b). Grazed grassland is forced in each grid-cell

by the prescribed N fertilizer application density and the historical gridded grazing-ruminant density. Stocking rate variability, starting, stopping and resumption of grazing periods during the growing season are simulated by ORCHIDEE-GM v3.1 (Vuichard et al., 2007; Chang et al., 2015b).

### 2.4 Grassland management intensity and historical changes

Herrero et al. (2013) established a global livestock production dataset containing a high-resolution (8 km × 8 km) gridded map of grass-biomass use for the year 2000. In this study, this dataset is extrapolated backwards in time from 2012 to 1901 to constrain the grass-biomass consumption in ORCHIDEE GM v3.1 in order to establish historical changes in the spatial distribution of grassland

management intensity.  Assuming that grass-biomass use for grid cell $k$ in country $j$ and year $m$





($GBU_{m,j,k}$ in kg dry matter (DM) per year) varies proportionally with the total ME requirement of domestic ruminants in each country, $GBU_{m,j,k}$ can be calculated from its value during the year 2000 given by Herrero et al. (2013), according to :

$$GBU_{m,j,k} = GBU_{2000,j,k} \times \frac{I_{m,j}}{I_{2000,j}} \qquad (1)$$

where $I_{m,j}$ and $I_{2000,j}$ are ME index (unitless) values for country $j$ in year $m$ and year 2000 respectively and given by:

$$I_{m,j} = \frac{ME_{m,j}}{ME_{ref,j}} \qquad (2)$$

where $ME_{m,j}$ is the total ME requirement by all ruminant (including cattle, sheep and goats) in country $j$ in year $m$; and $ME_{ref,j}$ is the total ME requirement by all ruminants in country $j$ in the reference year

2000 (see in Supplementary Information Text S3 for details).

ORCHIDEE-GM v3.1 simulates the potential (maximal) cut biomass from mown grasslands ($Y_{mown}$, unit: kg DM m$^{-2}$ yr$^{-1}$ from mown grassland) and the potential grazed biomass per unit area ($Y_{grazed}$, unit: kg DM m$^{-2}$ yr$^{-1}$ from grazed grassland) in each grid-cell. $Y_{grazed}$ is calculated as being driven by the

historical maps of grazing-ruminant density (see above and Supplementary Information Text S3). To avoid economically implausible stocking rates, we set a minimum grazing-ruminant density of 0.2 LU ha$^{-1}$. $Y_{grazed}$ is usually lower than $Y_{mown}$ in temperate grasslands, due to the lower herbage-use efficiency of grazing simulated by ORCHIDEE-GM (Chang et al., 2015b). However, in some arid regions, the grass biomass does not grow enough during the season to trigger harvest, i.e., it does not reach the

threshold in the model at which farmers are assumed to decide to cut grass for feeding forage to animals (see Chang et al., 2015b), so that $Y_{grazed}$ can become larger than $Y_{mown}$ (Fig. S1). The following set of rules was used to reconstruct historical changes in grassland management intensity, based on NPP simulated by ORCHIDEE-GM v3.1:

Rule-1: for each grid-cell and year, the total biomass removed by either grazing and cutting must be equal to the grass-biomass use, $GBU_{m,j,k}$ ;

Rule-2: grazing management prioritizes fulfilling $GBU_{m,j,k}$;

Rule-3: if the potential biomass consumption from grazing ($Y_{grazed}$) is not high enough to fulfil $GBU_{m,j,k}$, a combination of grazing and mowing management is undertaken.

Thus, for grid-cell $k$ in year $m$, the *minimum* fraction of grazed ($f_{grazed,m,k}$), the *minimum* fraction of mown ($f_{mown,m,k}$) and the *maximum* fraction of unmanaged grassland ($f_{unmanaged,m,k}$) are calculated with

the following equations (definitions of *minimum* and *maximum* in this context are given below).





If $A_{grass,m,k} \times Y_{grazed,m,k} > GBU_{m,j,k}$ , then:

$$f_{grazed,m,k} = \frac{GBU_{m,j,k}}{A_{grass,m,k} \times Y_{grazed,m,k}} \qquad (3)$$

$$f_{mown,m,k} = 0 \qquad (4)$$

$$f_{unmanaged,m,k} = 1 - f_{grazed,m,k} \qquad (5)$$

where $A_{grass,m,k}$ (unit: m$^2$) is the grassland area for grid-cell $k$ in year $m$ of the series of historic land-cover change maps (Supplementary Information Text S3).

If $A_{grass,m,k} \times Y_{grazed,m,k} < GBU_{m,j,k}$ , and $A_{grass,m,k} \times Y_{mown,m,k} > GBU_{m,j,k}$ , then:

$$f_{grazed,m,k} \times A_{grass,m,k} \times Y_{grazed,m,k} + f_{mown,m,k} \times A_{grass,m,k} \times Y_{mown,m,k} = GBU_{m,j,k} \qquad (6)$$

$$f_{grazed,m,k} + f_{mown,m,k} = 1 \qquad (7)$$

$$f_{unmanaged,m,k} = 0 \qquad (8)$$

If $GBU_{m,j,k}$ cannot be fulfilled by any combination of modelled $Y_{grazed}$ and $Y_{mown}$, we diagnose a *modelled grass-biomass production deficit* and apply the following equations :

if $Y_{grazed} > Y_{mown}$, then $f_{grazed,m,k} = 1$, $f_{mown,m,k} = 0$, and $f_{unmanaged,m,k} = 0$ $\qquad (9)$

if $Y_{grazed} < Y_{mown}$, then $f_{mown,m,k} = 1$, $f_{grazed,m,k} = 0$, and $f_{unmanaged,m,k} = 0$ $\qquad (10)$

This set of equations is valid for a mosaic of different types of grasslands in each grid-cell, some managed (grazed and/or mown) and some remaining unmanaged. In reality 1) farm owners could increase the mown fraction to produce more forage which corresponds approximately to the *mixed and landless* systems of Bouwman et al., (2005); and 2) animals could migrate a long way across grazed and unmanaged fractions (as they do in real rangelands) and only select the most digestible grass in pastoral systems, which corresponds to *extensively grazed* grasslands. Yet, given the approximations made in this study, $f_{grazed,m,k}$ and $f_{mown,m,k}$ represent the *minimum* fractions of grazed/mown grasslands rather than the actual fractions, and on the other hand $f_{unmanaged,m,k}$ corresponds to a *maximum* fraction of unmanaged grasslands since both *mixed and land less* and *extensive grazing* are not modelled.

### 2.5 Modelled productivity





Two simulation experiments were performed to simulate the global distribution of grassland productivity from 1901 to 2012. The experimental design aims to compare GPP and NPP of managed and unmanaged grasslands, with observation derived datasets. The unmanaged simulation is hereafter expressed as Sim-GU, and the managed one is Sim-GM: it includes all forcing data and Eqss (3-8) to

calculate the variable fractions of grazed, mown and unmanaged grassland in each grid-cell.

### 2.6 Datasets for model evaluation

#### 2.6.1 Grassland NPP observation database

Net primary productivity (NPP), including aboveground and belowground plant organs, represents the net flux of carbon from the atmosphere into live plant tissues (over one year in this study). NPP is a crucial variable in vegetation models and it is essential that this variable is properly validated. High quality measurements of grassland NPP are scarce, partly due to the difficulty of measuring some NPP

components such as fine-root production (Scurlock et al., 1999, 2002). An updated version of the Luyssaert et al. (2007) database comprising non-forest biomes (Campioli et al., 2015) was used here. This database attributes a flag indicating *managed* or *un-managed* to each site, and provides mean annual temperature, annual precipitation and downwelling solar radiation based on site measurements from the literature, CRU database (Mitchell and Jones, 2005), MARS database

(http://mars.jrc.ec.europa.eu/mars/About-us/AGRI4CAST/Data-distribution/AGRI4CAST-Interpolated-Meteorological-Data) or WorldClim database (Hijmans et al., 2005). Two additional datasets used in this study present NPP measurements from 30 sites across China (Zeng et al., 2015; Y. Bai, personal communication, 2015). These data include aboveground and belowground NPP observations at fenced (i.e., *unmanaged*) and unfenced (i.e., *managed*) grassland for each site. In total,

we selected 257 NPP observations (NPP of whole plant) with separated aboveground and belowground NPP from 113 sites all over the world (including grassland, and savanna; Fig. S2). Duplicate observations from the same site-year were averaged and considered as a single observation. NPP measurements with different management (managed or un-managed) at the same site were considered as two identical observations. In total, 214 grassland NPP measurements were compared to the

simulation of ORCHIDEE-GM v3.1 for the grid-cell corresponding to each site and for the time period of observation.

#### 2.6.2 Grassland GPP from MODIS products

The MOD17A3 dataset (version 55; Zhao et al., 2005; 2010) — a MODIS (the Moderate Resolution Imaging Spectroradiometer) product on vegetation production — provides the seasonal and annual GPP data at a spatial resolution of 1 km from 2000 to 2013. The MOD17 algorithm (Heinsch et al., 2003) uses the MODIS Land Cover Type product (MOD12Q1) as input employing Boston University's UMD classification scheme. To obtain the grassland GPP from the MOD17 dataset, we first extract the

MOD17 GPP at 1 km resolution over grassland grids in the MOD12Q1 dataset. Here, the grassland in





the MOD12Q1 dataset includes the 'open shrubland', 'savanna', and 'grassland' in the UMD classification scheme. The extracted annual and seasonal MOD17 GPP was then averaged and aggregated to $0.5^o \times 0.5^o$ spatial resolution to be comparable to model output. The grassland GPP simulated by ORCHIDEE-GM v3.1 was evaluated against the MOD17 GPP for the spatial pattern

(annual mean GPP), the seasonal cycle, and the interannual variability (IAV) (detrended time-series from 2000 to 2013).

### 2.6.3 Sun-induced chlorophyll fluorescence (SIF) data

Space-based observations of sun-induced chlorophyll fluorescence (SIF) provide a time-resolved measurement of a proxy of photosynthesis (Guanter et al., 2014). Similar to the MPI-BGC data-driven GPP product (Jung et al., 2011), SIF values exhibit a linear relationship ($r^2 = 0.79$) with monthly tower GPP at grassland sites in western Europe (Guanter et al., 2014). Compared to MODIS EVI (MOD13C2 products), SIF observations drop to zero during the non-growing season, thus providing a less clear

signal of photosynthetic activity (Guanter et al., 2014) than other vegetation indices based on visible and near-infrared reflectances. SIF also provides a better seasonal agreement with GPP from flux towers as compared to vegetation indices (Joiner et al., 2014). Guanter et al. (2014) showed that SIF data tend to better capture spatial hotspots of GPP (e.g., the US corn belt) than MODIS products.

A global SIF dataset was produced using spectra from the Global Ozone Monitoring Experiment-2 (GOME-2) instrument onboard the MetOp-A platform (Joiner et al., 2013). In this study, daily SIF retrievals from 2007 to 2012 are aggregated to monthly values (Version 26 (V26), Level 3 products with the spatial resolution of $0.5° \times 0.5°$) and averaged for each month to produce a mean seasonal variation related to photosynthetic activity. The seasonal variation of SIF is normalized (with the mean

value = 1) to evaluate the seasonality of grassland GPP simulated by ORCHIDEE-GM v3.1. In the GOME-2 SIF pixels that have a ground footprint of ~40km by 80km at nadir view during the time period examined, different PFTs can co-exist in the same grid-cell with different phenologies — this could bias the seasonality of grassland GPP. To reduce the contamination of SIF by non-grassland PFTs, we restrict the model-data comparison to grassland-dominated grid-cells, defined as those with

grassland cover in the MOD12Q1 dataset (Sect. 2.5.2) is larger than 50%.

Furthermore, SIF-GPP is calculated by SIF-GPP = −0.10 + 3.72 × SIF (V14) as given by Guanter et al. (2014) based on comparisons with cropland and grassland flux tower sites in the northern hemisphere at middle latitudes. However, SIF data V26 used in this study differs somewhat in magnitude from

V14 used by Guanter et al. (2014). To obtain the SIF-GPP linear model for SIF V26, we performed a linear regression between SIF V26 and SIF V14 monthly data over the fourteen gridcells that encompass the flux towers used in Table S1 of Guanter et al. (2014) and for the same time period. The resultant relationship obtained, is SIF-V14 = 1.25 × SIF-V26 ($r = 0.96$). The linear model SIF-GPP = -0.1 + 4.65 × SIF (V26) is used to calculate SIF-GPP in this study.



### 2.7 Model-data agreement metrics

Model-data agreement of NPP (modelled NPP vs. $NPP_{obs}$) and GPP (modelled GPP vs. MODIS-GPP) was assessed using Pearson's product-moment correlation coefficients and root mean squared errors (RMSE). The Pearson's product-moment correlation coefficient (r) describes the proportion of the total variance in the observed data that can be explained by the model, given by:

$$r = \frac{\sum_{i=1}^{n}(P_i - \overline{P})(O_i - \overline{O})}{\sqrt{\sum_{i=1}^{n}(P_i - \overline{P})^2}\sqrt{\sum_{i=1}^{n}(O_i - \overline{O})^2}} \qquad (11)$$

Where $P_i$ is modelled data, $O_i$ is observed data, $\overline{P}$ is modelled mean, $\overline{O}$ is observed mean, and n is the sample size. The RMSE is a measure of model accuracy reporting the mean difference between the modelled and observed fluxes, expressed as:

$$RMSE = \sqrt{\frac{\sum_{i=1}^{n}(P_i - O_i)^2}{n}} \qquad (12)$$

where $P_i$ is modelled data, $O_i$ is observed data, and n is the sample size.

$r$ is used to assess the model-data agreement of GPP for the spatial pattern and the IAV (modelled GPP vs. MODIS-GPP), and seasonality (modelled GPP vs. MODIS-GPP and SIF data).

### 3 Results

### 3.1 Maps of grassland management intensity

Figure 1 shows the minimum fractions of mown and grazed grasslands, and the maximum fraction of unmanaged out of total grassland ($f_{mown}$, $f_{grazed}$, and $f_{unmanaged}$ respectively; Sect. 2.4) in the year 2000. Grazed grasslands comprise most of the managed grasslands in the maps (Fig. 1b). Significant fractions of mown grasslands are only found in regions with high ruminant stocking density such as eastern China, India, eastern Europe and eastern United States, where $Y_{grazed}$ cannot fulfil the grass-biomass demand (Fig. 1a). Using the FAO-defined regions (see caption to Fig. 3), the largest fractions of managed grasslands are modelled in regions of high ruminant density (Fig. S3) such as in Eastern Europe with a mean fraction of 89 ± 17% (the mean being the average fraction of mown and grazed grasslands over all the grid-cells in this region and the standard deviation being taken from differences between grid-cells), South Asia (82 ± 23%), western Europe (55 ± 33%), and North America (49 ± 35%). Lower managed grasslands fractions are modeled in Oceania (42 ± 35%), Latin America and the Caribbean (LAC, 40 ± 27%), the Russian Federation (40 ± 38%), and sub-Saharan Africa (SSA, 38 ± 35%).





In some grid-cells, the simulated grassland productivity is not sufficient to fulfil the grass-biomass use given by Herrero et al. (2013; Fig. 1d). Of the 2.4 billion tonnes of grass-biomass use (in dry matter) given by Herrero et al., 18% cannot be fulfilled by the productivity simulated by ORCHIDEE-GM v3.1. This translates into a modelled grass-biomass production deficit of 0.42 billion tonnes (Table 2). Out of all regions, the largest modelled production deficit ($f_{global}$ in Table 3) is found in South Asia (50%). This South Asian deficit is predominantly in India (36%) and Pakistan (9%). Other regions with a biomass production deficit are the Near East and North Africa (NENA; 16%) and sub-Saharan Africa (SSA 11%). Overall, 24% of the global production deficit comes from regions with dry climate and low NPP (less than 50 g C m$^{-2}$yr$^{-1}$), and 10% of it comes from regions with low grassland cover (less than 10% of total land cover). The causes of this grass-biomass production deficit diagnosed by ORCHIDEE-GM are discussed in Sect. 4.2.

### 3.2 Modelled productivity

Figure 2 shows the grassland productivity (NPP) from the simulation Sim-GM (Fig. 2a), and the NPP differences between Sim-GM and Sim-GU (Fig. 2b). The effect of including management does not produce a big difference in simulated NPP, which has similar patterns between Sim-GM and Sim-GU in most regions (Fig. 2b). Nevertheless, there are significant differences of NPP due to management in the central United States, Europe, south China, South Korea, south Japan, and south Brazil where N fertilizer additions (Table S3) cause a higher productivity (Fig. 2c and Fig. 3).

Figure 3 displays the NPP per unit area, and the production ($Prod$ = NPP × grassland area) of each type of grassland for ten FAO-defined regions and the globe in Sim-GM. Even when grassland management is included, the production of unmanaged grassland ($Prod_{unmanaged}$) still comprises 66% of the total production ($Prod_{total}$) in the 1990s. The production of grazed grasslands ($Prod_{grazed}$) accounts for 31% of $Prod_{total}$, while the production of mown grasslands ($Prod_{mown}$) is only 3%, given the small area under this management practice (Fig. 3). Mown grasslands only contribute to production in the regions where climate conditions and fertilizers maintain a high NPP, and $Y_{grazed}$ is not enough to fulfil the animal requirement, which triggers the harvest practice in Equations (6-8). As a result, Western and Eastern Europe and South Asia have higher $Prod_{mown}$ fractions.

### 3.3 Historical changes in the area and productivity of managed grassland

The global minimum area of managed grassland ($A_{managed-gm}$) is of 5.1 × 10$^6$ km$^2$ in 1901 and increased to 11 × 10$^6$ km$^2$ in 2000 (Table 3; Fig. 4) — an increase of 116% during the 20$^{th}$ century. This expansion of managed grasslands is mainly explained by the increase in the area of grazed lands (+5.3 × 10$^6$ km$^2$) while mown grassland increased only marginally (+0.6 × 10$^6$ km$^2$). The largest extension of $A_{managed-gm}$ (+1.6 × 10$^6$ km$^2$) is found in Latin America and the Caribbean (LAC), and Sub-Saharan Africa (SSA; Fig. 4). The regions with the largest relative expansion of managed grasslands (as a





percentage of 1901 areas) are Sub-Saharan Africa (+260%) and East and Southeast Asia (E & SE Asia; +245%), a region where the number of domestic ruminants ($N_{ruminant}$) increased by nearly a factor of five. Only small increases of $A_{managed-gm}$ were modeled in Western Europe (+37 × $10^3$ km$^2$; i.e., 7.7%) and Eastern Europe (+30 × $10^3$ km$^2$; i.e., 8.1%), despite an increase of $N_{ruminant}$ by a factor of 1.5 in

Western Europe (+27 × $10^6$ LU), and of 1.4 in Eastern Europe (+5 × $10^6$ LU). This means that livestock production intensified in those two regions, first by giving crop feedstock given to animals (Bouwman et al., 2005) and second through the optimization of forage harvesting and grazing to feed higher animal-stocking densities. Note that the animal density in Eastern and Western Europe peaked at 123 × $10^6$ LU near 1990, and has declined by 29% since then.

In addition to the extension of managed grassland areas since 1901, the ratio of mown-to-grazed grasslands ($R_{mown-to-grazed}$) has increased from 6% in 1901 to 8% in 2000. The largest increases in $R_{mown-to-grazed}$ are found in East and Southeast Asia (ESA; by a factor of 4.6), Near East and North Africa (NENA; by a factor of 4.2), and Oceania (by a factor of 3.2). By contrast, $R_{mown-to-grazed}$ increased less in

Russia (by a factor of 1.3) and decreased in Western and Eastern Europe (by factors of 0.7 and 0.8 respectively). It is noteworthy that high $R_{mown-to-grazed}$ values are modelled in NENA, ESA and Oceania for the 1960s, 1970s and 1980s when the number of ruminants increased and was higher than today (Fig. 4). For other regions, the $R_{mown-to-grazed}$ increased by a factor ranging between 1.3 and 2.0 from 1901 to 2000.

The global mean potential productivity of mown grassland ($Y_{mown}$) increased by 55% from 312 g DM m$^{-2}$ yr$^{-1}$ for 1900s to 484 g DM m$^{-2}$ yr$^{-1}$ for the 1990s, while that of grazed grassland $Y_{grazed}$ increased by 33%, from 117 g DM m$^{-2}$ yr$^{-1}$ for the 1900s to 156 g DM m$^{-2}$ yr$^{-1}$ for the 1990s (Table 4). During the last century, $Y_{mown}$ increased by more than 60% in most regions except in Latin America and the

Caribbean (18%) and North America (43%), while the increase of $Y_{grazed}$ ranged from 22% in Sub-Saharan Africa and 67% in Eastern Europe (Table 3).

### 3.4 Evaluation of modelled NPP against observed NPP

Figure 5 shows the comparison between site-scale NPP observations ($NPP_{obs}$) and the model results at the corresponding grid-cells. The modelled NPP is positively correlated with $NPP_{obs}$ across 113 sites but the value of the correlation coefficient is low ($r = 0.33 - 0.36$, $p < 0.01$). With calibrated parameters and management, the RMSE of $NPP_{Sim-GU}$ is 397 g C m$^{-2}$yr$^{-1}$ ($r = 0.33$, $p < 0.01$). When using NPP of

mown grassland instead of unmanaged grassland as modelled NPP for managed sites, $r$ increases a little but the RMSE is not changed (396 g C m$^{-2}$ yr$^{-1}$; $r = 0.36$, $p < 0.01$; Fig. 5b). Figure 5c presents box-and-whiskers plot of the observed and modelled annual whole-plant NPP, aboveground NPP and belowground NPP. The mean value and range of modelled whole plant NPP are both higher than those of $NPP_{obs}$. The NPP overestimation by the model is mainly due to a too-high aboveground NPP, while

belowground NPP is similar for its mean or even lower for its median, than belowground $NPP_{obs}$.



### 3.5 Evaluation of modelled GPP against MODIS-GPP for annual mean and interannual variability

At global scale, MODIS-GPP gives a mean grassland GPP of 537 g C m$^{-2}$ yr$^{-1}$, and ORCHIDEE-GM v3.1 simulates a mean value of 813 g C m$^{-2}$ yr$^{-1}$ for $GPP_{Sim-GU}$ and 791 g C m$^{-2}$ yr$^{-1}$ for $GPP_{Sim-GM}$, ≈ 50% higher than MODIS-GPP. $GPP_{Sim-GU}$ is very similar to $GPP_{Sim-GM}$, indicating that management does not explain the higher values than MODIS. A higher modelled GPP than MODIS is found for all latitude bands especially in boreal (50$^{o}$N – 80$^{o}$N) and tropical regions (20$^{o}$S – 20$^{o}$N; Fig. 6). We performed

linear regressions between MODIS-GPP and modelled GPP for all the 0.5° grid-cells with grassland covering more than 20% of total land (i.e., grassland is not a trivial land cover in that grid-cell) in the MOD12Q1 dataset. The slope of the regression between $GPP_{Sim-GU}$ and MODIS-GPP is 1.01, and the correlation coefficient ($r_{spatial}$) is 0.85 (Table 4). This suggests that the spatial pattern of MODIS-GPP is similar to that of $GPP_{Sim-GU}$. Similar slope and $r_{spatial}$ values resulted from $GPP_{Sim-GM}$, which indicates

that the model comparison against MODIS-GPP is not improved by including management (Table 4).

With 14 years of global coverage (2000 – 2013), the MODIS-GPP product can also be used to evaluate the interannual variability (IAV) of GPP. The temporal correlation coefficient between the detrended time-series of global $GPP_{Sim-GU}$ and MODIS-GPP was found to be high ($r_{IAV-global}$ = 0.88, p < 0.01;

Table 4) Given the similar $r_{IAV}$ between both $GPP_{Sim-GU}$ and $GPP_{Sim-GM}$, and MODIS-GPP, one can conclude that management does not change the IAV of GPP in the model. This is because IAV is mainly driven by climate, whereas management responds to trends (Chang et al., 2015c). Within the grid-cells covered by grass over more than 20% of total land in MOD12Q1, significant positive interannual correlations between $GPP_{Sim-GM}$ and MODIS-GPP were found for 40% of the grid-cells (i.e.,

42% of the grassland area), except in some tundra areas of Siberia and North America, grassland on the Qinghai-Tibet Plateau, and savannah in Sub-Saharan Africa (Fig. 8).

We use the coefficient of variation (CV) to compare the magnitude of the IAV of GPP between $GPP_{Sim-GM}$ and MODIS-GPP (Fig 9). In general, the average CV of $GPP_{Sim-GM}$ (37%) is a factor of three

larger than that of MODIS-GPP (13%). Relatively high values of CV (over 10%) are consistently found both in ORCHIDEE-GM and MODIS for semi-arid regions including western United States, central Asia, eastern Brazil, the Sahel region, southern Africa, and central Australia. But the CV value of $GPP_{Sim-GM}$ for those regions (> 30%) is higher than the one of MODIS-GPP. However, in most of the Northern Hemisphere tundra, MODIS-GPP gives a higher CV than $GPP_{Sim-GM}$.

### 3.6 Evaluation of modelled seasonal cycle of GPP against MODIS-GPP and GOME-2 SIF products

Figure 10 compares the normalized seasonal variation of GPP ($GPP_{Sim-GM}$), MODIS-GPP, and SIF for

five latitude bands and the globe. The seasonal variations of $GPP_{Sim-GU}$ are almost the same as for




$GPP_{Sim-GM}$ (Table 5) indicating management does not change the grass phenology significantly. Similar mean seasonal variations of grassland productivity are found between modelled GPP, MODIS-GPP and SIF ($r_{seasonal}$ range from 0.55 to 0.90; Fig. 10). Compared to both MODIS-GPP and SIF data, ORCHIDEE-GM v3.1 captures the seasonal variation of productivity in boreal and temperate regions

of the Northern Hemisphere well ($r_{seasonal} > 0.8$; Table 5; Figs. 11a and b). In the band from 60°S to 30°N, significant positive $r_{seasonal}$ correlations are found both with MODIS-GPP and SIF (Fig. 11a and b). Non-significant or negative $r_{seasonal}$ values occur however in eastern Africa, in some regions of South America, and in central Australia (Fig. 11), which cause the low average $r_{seasonal}$ for the corresponding latitude bands (Table 5). However, note that the $r_{seasonal}$ between the two remote sensing

GPP related products is relatively low for grassland between 60°S and 30°N (Fig. 11c), particularly between 0-60°S (Table 5).

We further compare the maximum monthly GPP ($GPP_{max}$) from ORCHIDEE-GM v3.1, MODIS-GPP and SIF-GPP, to investigate whether the model can capture the $GPP_{max}$ and its spatial gradient. Only

$GPP_{max}$ from Sim-GM is shown, because $GPP_{Sim-GM}$ is very similar to $GPP_{Sim-GU}$ in its spatial pattern. ORCHIDEE-GM v3.1 tends to produce higher $GPP_{max}$ than MODIS-GPP and SIF-GPP all over the world (Fig. 12), but especially in tundra. It is worth noting that $GPP_{max}$ from Sim-GM is more close to SIF-GPP in magnitude; SIF-GPP has generally higher $GPP_{max}$ values than MODIS-GPP in temperate and sub-tropical regions. When excluding northern tundra, $GPP_{max}$ from Sim-GM shows a similar

spatial gradient to that from SIF-GPP ($r = 0.55$, $p < 0.01$) and MODIS-GPP ($r = 0.59$, $p < 0.01$), while $GPP_{max}$ from SIF-GPP and MODIS-GPP are very similar ($r = 0.85$, $p < 0.01$).

## 4 Discussion

### 4.1 Managed area of grassland and management intensity: comparison with previous estimates

The area of managed grasslands obtained in this study is lower than the *pasture area* of HYDE 3.1 ($A_{pasture-hyde}$, Klein Goldewijk et al., 2011; Table 3), except in Eastern Europe for the year 2000. $A_{pasture-hyde}$ is 3.8 times larger than $A_{managed-gm}$ in the year 1901 and 3.0 times larger in the year 2000. The

difference comes from the method used for estimating managed areas between Klein Goldewijk et al. (2011) and this study. $A_{pasture-hyde}$ in Klein Goldewijk et al. (2011) was estimated simply from population density and the country-level per capita use of pasture derived from the FAO statistics (FAO, 2008). In this study, $A_{managed-gm}$ including the minimum area of mown plus grazed grasslands, is constrained by grass-biomass use data (i.e., requirement of biomass for animals) and the simulated

grassland productivity (i.e., supply of biomass to animals). In fact, the actual (real-world) managed grassland area could be larger than $A_{managed-gm}$ in regions where grasslands are not strictly un-managed, i.e., not fully occupied by $A_{managed-gm}$ in the management intensity maps (i.e., $f_{unmanaged} > 0$; Fig. 1c). In pastoral systems such as open rangeland and mountain areas, animals keep moving to search for the most digestible grass. Tracts of grasslands can be grazed for a short period, with only a small part of

the annual grass productivity being digested (i.e., very low herbage-use efficiency). This type of





grassland could be recognized as extensively grazed grassland, whereas it is considered as unmanaged in this study. For example, lower herbage-use efficiency than that simulated in this study (Fig. 13) could be expected in open rangeland of central Asia, the Russia federation, sub-Saharan Africa, Brazil and Australia, and in the mountains of southwest China and the European Alps. Reclassifying these areas would result in a larger area of extensively managed grassland. Few studies reported the herbage-use efficiency of managed grassland. One exception is the network of European eddy-covariance flux sites. For these sites the average herbage-use efficiency (expressed as forage defoliated as a propotion of GPP) is 7.1% ± 6.1% for grazed sites, and 13.3% ± 6.4% for mown sites (J-F. Soussana, personal communication, 2015); a similar range, between 2% and 20% is simulated in this study (Fig. 13).

The time evolution of $A_{managed\text{-}gm}$ since 1901 in this study is arguably more realistic than HYDE because it considers changes in animal stocking density from statistics and the evolution in per-head use of pasture. $A_{managed\text{-}gm}$ takes into account 1) changes in grass-biomass requirement considering both ruminant numbers and meat/milk productivity (Supplementary Information Text S3; $N_{ruminant}$ in Table 3); 2) changes in grassland productivity driven by climate change, rising $CO_2$ concentration, and changes in N fertilization ($Y_{mown}$ and $Y_{grazed}$ in Table 3); and 3) changes in management types (mown and grazed grassland areas in Table 3 and Fig. 4). For example in intensively managed grasslands, an increase in ruminant stocking density causes a shift from grazed to mown grassland (globally and regionally, except in Europe; Table 3 and Fig. 4), because mown grassland provides more grass biomass than grazed grassland per unit of area (Fig. S1).

$A_{pasture\text{-}hyde}$ is consistent with country-specific pasture area censuses, and thus may be suitable for reconstructing land-cover, but it does not provide information about management intensity. $A_{managed\text{-}gm}$ and its split between mown, grazed and unmanaged fractions provides specific global distributions of *pasture* management intensity and its historical changes. However, there are several limitations, which may cause uncertainties in our maps of management intensity: 1) the grass fraction in ruminant diet has likely been changing during the last century, while due to the lack of information, we assumed that it was static in each region up to the year 2000; 2) technical development (such as ruminant breeding) are not considered, but may affect the feeding efficiency (meat/milk production per amount of feed) and thus feedback on the grass-biomass requirement; 3) the spatial distribution of ruminants was kept constant in our estimate, whereas it could have changed, depending on geographic changes in human population distribution; and 4) the results depend on the accuracy of NPP modeling in ORCHIDEE-GM. Despite these limitations, the maps of grassland management intensity provide new information for drawing up global estimates of management impact on biomass production and yields (Campioli et al., 2015) and for global vegetation models like ORCHIDEE-GM to enable simulations of carbon stocks and GHG budgets beyond simple tuning of grassland productivities (e.g., like in LPJmL; Bondeau et al., 2007) to account for management. These maps can also be tested in other DGVMs, or the same algorithm implemented in other models to give the management intensity consistent with simulated NPP.





### 4.2 Causes of regional grass-biomass production deficits

Grass-biomass production is constrained by the gridded biomass consumption for the year 2000
(Herrero et al., 2013). In some grid-cells, the gridded biomass consumption by year 2000 cannot be
fulfilled by the potential grass production simulated by ORCHIDEE-GM v3.1 (Fig. 1b). These
modelled grass-biomass production deficits could be due to several reasons:

- Land-cover maps used as input to ORCHIDEE-GM v3.1 do not represent grasslands well in
  the *mixed and landless* systems, and grasslands providing occasional feed to ruminant (e.g.,
  roadside, forest understory grazing land, and small patches). This failing could cause the
  model to miss a significant part of grass productivity in this study. For example, the largest
  modelled grass-biomass production deficit is found in India because the simulated grassland
  productivity is far from agreeing with the grass biomass use data. In this country, occasional
  feed may constitute an important fraction of ruminant diet (30% or 50% in *mixed and landless*
  or pastoral systems of south Asia from Bouwman et al., 2005), which is not represented by the
  land-cover maps used as input to ORCHIDEE-GM v3.1 and thus is not modelled.

- In arid regions such as Pakistan, Sudan, Iran, Egypt and in northwest China, grass can grow in
  places where the water table is near to the surface and groundwater resources are available
  (e.g., oases, riparian zones, lakes). However, ORCHIDEE-GM v3.1 is driven by gridded
  climate data and does not taken into account local topography-dependent water resources such
  as rivers and lakes, and thus is not being able to simulate local grass growing areas in arid
  regions.

- Grassland irrigation, though it is not as common as in cropland, is applied in arid regions such
  as Saudi Arabia, but is not considered by ORCHIDEE-GM v3.1.

- In some semi-arid open rangeland, ruminants may walk long distances to acquire enough
  grass. For example, in semi-arid sub-Saharan Africa, Uzbekistan and central Australia,
  animals usually keep moving in order to search for grass. This displacement of grazing
  animals from grass sources is not considered in the model.

- The grass fraction in ruminant diet is defined per region according to specific production
  systems. However, the grass fraction can differ within a region depending on local fodder crop
  production and grassland use. For example, the large numbers of ruminants in eastern China
  are mostly fed by grain and stovers (fibrous crop residues) instead of grass, because little
  grassland exists in that region.

### 4.3 Model performance: comparison of modelled and observed grassland productivity

In Sects. 3.4 and 3.5, the spatial patterns of modelled productivity (NPP or GPP) were compared with
observations ($NPP_{obs}$ or MODIS-GPP). ORCHIDEE-GM v3.1 did well at capturing the spatial pattern
of grassland productivity, with: i) high $r_{spatial}$ between modelled GPP and MODIS-GPP (Sect. 3.5); and
ii) modelled NPP extracted from global simulation showing significant correlation with site-level NPP





observation from 113 sites all over the world (Sect. 3.4). However, modelled annual GPP is higher than MODIS-GPP in all latitude bands (Fig. 6). It should be kept in mind that MODIS-GPP was diagnosed an 18% uncertainty due to climate forcing (Zhao et al., 2006). Besides, a low bias of MODIS-GPP for grasslands has been reported in a tallgrass prairie in the United States (Turner et al., 2006) and in an

alpine meadow on the Tibetan Plateau (Zhang et al., 2008), when compared to the GPP from flux-tower measurements. The underestimate of MODIS-GPP is mostly related to the low value of the maximum light-use efficiency parameters used in the MODIS-GPP algorithm (Turner et al., 2006; Zhang et al., 2008).

The relatively low $r$ value between modelled NPP and site-level $NPP_{obs}$ (r = 0.33 – 0.36, p < 0.01; Sect. 3.4) could be related to the fact that local climate, soil properties, topographic features are not considered in the model. For example, the $r$ between the site-level climate and that from the CRU+NCEP climate forcing data ($0.5^o \times 0.5^o$ resolution) are 0.96 for annual mean temperature, but only 0.86 for annual total precipitation and 0.86 for solar radiation. The relatively low correlation for

annual total precipitation may cause inaccuracy in the model simulations of productivity, because water availability could be a major factor limiting grass growth (e.g., in temperate regions, Le Houerou et al., 1988; Silvertown et al., 1994; Briggs and Knapp 1995; Knapp et al., 2001; Nippert et al., 2006; Harpole et al., 2007). Further, compared to observed aboveground and belowground NPP, a similar mean belowground NPP and an overestimation of mean aboveground NPP by ORCHIDEE-GM v3.1 is

found in Sect. 3.4, which suggests that 1) the model tends to overestimate aboveground NPP possibly due to overestimation of GPP (compared to MODIS-GPP), and 2) the model tends to overestimate the ratio of aboveground and belowground biomass allocation ($R_{above/below}$) compared to observation. This overestimation could be the result of nitrogen limitation on the carbon allocation scheme for grassland. For example, high nitrogen supply has been observed to increase $R_{above/below}$ (Aerts et al., 1991; Cotrufo

and Gorissen, 1997), while nitrogen limitation might cause it to decrease.  However, nitrogen limitation in grassland is not accounted for in ORCHIDEE-GM v3.1, which possibly leads to the model's overestimation of $R_{above/below}$. The model could be improved by incorporating the full nitrogen cycle.

For the seasonal cycle, we compared modelled GPP seasonality to both MODIS-GPP (MOD17A2

product) and GOME-2 SIF data. ORCHIDEE-GM v3.1 captures the seasonal variation of productivity in most regions where grassland is the dominant ecosystem (coverage > 50%), as shown by the high $r_{seasonal}$ between modelled GPP and MODIS-GPP (Fig. 11a) or SIF data (Fig. 11b). However, the model does not capture the seasonal amplitude of grassland productivity in some arid/semi-arid regions (e.g., southwest United States, and central Australia). In arid/semi-arid regions, grass productivity is

triggered by discrete precipitation events, and depends on the timing and magnitude of these pulses (Sala et al., 1982; Schwinning and Sala, 2004; Huxman et al., 2004). These precipitation pulses are infrequent, discrete, and not represented in a global climate re-analysis dataset such as CRU+NCEP used in our simulation. In particular, NCEP, like all climate models tends to produce "GCM drizzle" (Berg et al., 2010), i.e., too many frequent small rainfall events. This forcing uncertainty could be a

major obstacle for our model to capture the seasonality of productivity in these regions. In dry





grasslands, the dominant species could change during the season, but the resultant changes in SLA and $Vc_{max}25$ by different dominant species cannot be reflected in ORCHIDEE-GM v3.1. This within-season variability could be another reason for the model-data discrepancy in arid/semi-arid grassland seasonality. For the savanna of sub-Saharan Africa, eastern Africa and South America (Fig. 11), the

relatively low $r_{seasonal}$ could be result from the fact that the frequent fires are not simulated in the current version of the model used here. $GPP_{max}$, indicating the maximum photosynthetic activity within a year, could be another good indicator of plant seasonality. The comparison between $GPP_{max}$ from the model, the MODIS product and the SIF data (Sect. 3.6) reveals that: 1) ORCHIDEE-GM v3.1 greatly overestimates grassland $GPP_{max}$ of the boreal-tundra areas, but 2) the model generally captures the

same spatial gradient of $GPP_{max}$ than MODIS and SIF products. Tundra has its own specific characteristics and functioning traits associated with the extreme environment with a severely cold winter, frozen soil and short growing season, but is treated as normal C3 grassland in our model. Apart from the extreme environment, the low productivity of the tundra system is also attributed to low availability of nutrients and the slow nutrient cycle (e.g., Nilsson et al., 2002; Elser et al., 2007; Stark,

2007; Giesler et al., 2012). However, the nutrient cycles such as the nitrogen and phosphorus cycles are not considered in our model. Specific parameterization, and inclusion of the nutrient cycles for the tundra biome are required in the future to improve model performance.

ORCHIDEE-GM v3.1 captures the the same IAV of grassland GPP at global scale and in many regions

of the world (42% of global grassland area) than in the MODIS-GPP product. One exception where IAV is not in phase with MODIS-GPP is sub-Saharan Africa. Possible causes of this discrepancy are : 1) the frequent fires which affect the IAV of GPP, are not simulated in this study, 2) model biases in the IAV of soil moisture, which could affect the model performances for the productivity of semi-arid Africa, given its two-layer bucket hydrology; 3) the problems with MODIS-GPP dry areas, which may

degrade the model-data agreement.  The cold Qinghai-Tibet plateau is another region where the model does not capture the GPP IAV (Fig. 8), which could be due to shortcomings in the phenology parameterization and the snow scheme. For the Qinghai-Tibet plateau, the phenology of ORCHIDEE could be improved by a regional parameterization (Tan et al., 2010). Snow can exist for several months in Qinghai-Tibet plateau and in the high-latitude tundra of the Northern Hemisphere. The timing of

snowmelt will impact the grass phenology, while early spring soil moisture impacted by snow water storage may affect the grassland productivity. The single-bucket snowpack scheme (Chalita and Le Treut, 1994) in the current version of ORCHIDEE-GM may not represent the snow processes sufficiently accurately. The mechanistic intermediate-complexity snow scheme (ISBA-ES; Boone and Etchevers, 2001) implemented into ORCHIDEE-ES (Wang et al., 2013) may improve the model

performance in simulating grassland productivity. In boreal-tundra areas, again, the low model-data agreement in IAV could be due to the specific characteristics, functioning traits, and nutrient availability that are not well parameterized or accounted for in our model.

**5. Concluding remarks**



The carbon-water-energy land surface model ORCHIDEE-GM v3.1 is calibrated for C4 grass parameters representing photosynthetic ($Vc_{max}25$), and morphological plant traits ($SLA_{max}$) and includes specific parameterization of managed grasslands. The modelled distribution of grassland productivity on a 0.5° by 0.5° grid over the globe was evaluated against a series of observations from site-level NPP

measurements to global satellite-based products (MODIS-GPP and GOME2-SIF). Generally, ORCHIDEE-GM v3.1 captures the spatial pattern, seasonal cycle and IAV of grassland productivity at global scale, except in regions with either arid or cold climates (tundra) and high altitude mountains/plateaus. Because the major purpose of a global vegetation model like ORCHIDEE-GM is to simulate carbon, water, and energy fluxes at a large scale it uses a limited number of plant functional

types and generic equations. The model is not expected to accurately capture productivity variations everywhere. Thus we conclude that its current version,ORCHIDEE-GM v3.1 is suitable for use at simulating global grassland productivity.

In this study, we have derived the global gridded maps of grassland management intensity including

the minimum area of managed grassland with fraction of mown/grazed part, the grazing ruminant stocking density, and the density of the wild animal population.  The management intensity maps are built based on the assumption that grass-biomass production from managed grassland (simulated by ORCHIDEE-GM v3.1) in each grid-cell is just enough to satisfy the grass-biomass requirement by ruminants in the same grid (data derived from Herrero et al., 2013). Furthermore, the maps are

extended to cover the period 1901-2012, taking into account both the changes in grass-biomass requirement and supply. The evolution in grass-biomass requirement is determined by the ME-based ruminant numbers calculated in this study, while the changes in grass-biomass supply are simulated by ORCHIDEE-GM v3.1 considering variable drivers such as climate, $CO_2$ concentration, and N fertilization. Despite the multiple sources of uncertainty, these maps, to our knowledge for the first

time, provide global, time-dependent information on grassland management intensity. Global vegetation models such as ORCHIDEE-GM, containing an explicit representation of grassland management, are now able to use these maps to make a more accurate estimates of global carbon and GHG budgets.





**Acknowledgement.** We gratefully acknowledge funding from the European Union Seventh Framework Programme FP7/2007–2013 under grant N° 603864 (HELIX). P.C. and S.Pe acknowledge support from the ERC Synergy grant ERC-2013-SyG-610028 IMBALANCE-P. M.C. is a Postdoctoral Fellow of the Research Foundation – Flanders (FWO). C.Y. is supported by the European

5   Commission-funded project LUC4C (Grant N° 603542). T.W. is funded by European Union FP7-ENV project PAGE21 (Grant N° 282700). We thank the EC-JRC-MARS dataset (© European Union, 2011-2014) created by MeteoConsult based on ECWMF (European Centre for Medium Range Weather Forecasts) model outputs and a reanalysis of ERA-Interim. We greatly thank Dr. John Gash for his effort on English language editing.





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





**Figure legend**

Figure 1. (a) Mown, (b) grazed, and (c) unmanaged fraction of global grassland, and (d) modelled grass-biomass production deficit of 2000. Modelled grass-biomass production deficit indicates the simulated grassland productivity in the grid cells is not sufficient to fulfill the grass-biomass use given by Herrero et al. (2013), and is expressed with units of g dry matter (DM) per $m^2$ of total land area in each grid cell.

Figure 2. Modelled mean grassland NPP for the period 1990-1999 from the simulation experiments Sim-GM (a), and the NPP differences (b) between Sim-GM and Sim-GU. See Sect. 2.5 for the details of the experiments.

Figure 3. Productivities per unit area (height of each rectangle) and grassland areas (width of each rectangle) of the different types of grassland (mown, grazed, and unmanaged grassland) by FAO-defined regions and global total. Areas in the graph shows the production of each grassland type. Productivities and grassland areas are averaged for 1991-2000 from experiment Sim-GM. The FAO-defined regions (from top-left) are North America, Russian Federation, Western Europe, Eastern Europe, Near East & North Africa (NENA), East & Southeast Asia, Oceania, South Asia, Latin America and the Caribbean (LAC), Sub-Saharan Africa (SSA).

Figure 4. Historic changes in the area of managed/unmanaged grassland, and in the ruminant numbers for 1901 and 2012 by regions and global total. See caption to Fig. 3 for expansion of FAO-defined regions.

Figure 5. Comparison between site-observations of whole plant NPP ($NPP_{obs}$) and modelled NPP from experiments: (a) Sim-GU or (b) Sim-GM, and (c) box-and-whisker plot of the observed and modelled annual whole-plant NPP, aboveground NPP and belowground NPP. In subplot (a) and (b), grassland sites in different Köppen climate zones are specified by different colours. The Köppen climate zones are classified based on Peel et al. (2007) using climate data from WorldClim (http://www.worldclim.org/). In subplot (c), NPPs from different experiments are specified by different colours, and the 'whisker' indicates the cross-measurement (total 214 measurements) uncertainty.

Figure 6. Comparison between MODIS-GPP and modelled GPP from two experiments, by latitude band. $GPP_{Sim-GU}$ (green line) is nearly fully covered by $GPP_{Sim-GM}$ (red line). The uncertainty of MODIS-GPP comes from the reported relative error term driven by NASA's Data Assimilation Office (DAO) reanalysis datasets (Zhao et al., 2006). The uncertainty of modelled GPP is the standard deviation of interannual variation of grassland GPP in each band for the period 2000-2013. To make the figure less complicated and readable, only the uncertainty of $GPP_{Sim-GM}$ is presented as an example.

Figure 7. Comparison between MODIS-GPP and modelled GPP from two experiments at the resolution of $0.5^o \times 0.5^o$.

Figure 8. Spatial distribution of $r_{IAV}$ between MODIS-GPP and $GPP_{Sim-GM}$. $r_{IAV}$ is the correlation coefficient between detrended time-series of modelled and MODIS-GPP from 2000 to 2013. This figure only shows the $r_{IAV}$ for grid-cells with grassland covering more than 20% of total land in the MOD12Q1 dataset. Grey colour indicates insignificant or negative $r_{IAV}$ ($p > 0.05$ or r $<$ 0); and yellow-to-red indicate significant positive $r_{IAV}$ with increasing value (r $>$ 0 and p $<$ 0.05).

Figure 9. Coefficient of variation (CV) of (a) MODIS-GPP and (b) $GPP_{Sim-GM}$.

Figure 10. The normalized seasonal variation of modelled GPP ($GPP_{Sim-GM}$), MODIS-GPP, and SIF for five latitude bands (a – e) and (f) global average.

Figure 11. Spatial distribution of $r_{seasonal}$ between (a) SIF data and $GPP_{Sim-GM}$, (b) MODIS-GPP and $GPP_{Sim-GM}$, and (c) MODIS-GPP and SIF data respectively. $r_{seasonal}$ is the correlation coefficient between mean seasonal cycle of modelled GPP, MODIS-GPP and SIF data from 2008 to 2012. This figure only shows the $r_{seasonal}$ for grid-cells with grassland covering more than 50% of total land in the MOD12Q1 dataset. Grey colour indicates insignificant or negative $r_{seasonal}$ ($p > 0.05$ or r $<$ 0); and yellow-to-red indicate significant positive $r_{seasonal}$ with increasing value (r $>$ 0 and p $<$ 0.05).

Figure 12. The maximum monthly GPP ($GPP_{max}$) from (a) ORCHIDEE-GM v3.1 ($GPP_{Sim-GM}$), (b) MODIS-GPP and (c) GPP derived from SIF Version 26 (SIF-GPP). Data are monthly average for the period 2008 - 2012.



Figure 13. Average herbage-use efficiency over managed grassland (grazed plus mown) in 2000-2009 simulated by ORCHIDEE-GM v3.1. Herbage use efficiency (Hodgson, 1979) is defined as the forage removed expressed as a proportion of herbage growth. In this study, the forage removed is modelled annual grass biomass use including $Y_{grazed}$ and $Y_{mown}$, and herbage growth is modeled annual grass GPP.





**Tables**

Table 1. The forcing data for different types of grassland in the simulation.

| Forcing data | Grassland types | | |
|---|---|---|---|
| | Unmanaged grassland | Mown grassland | Grazed grassland |
| Atmospheric $CO_2$ concentration | Yes | Yes | Yes |
| Climate forcings | Yes | Yes | Yes |
| Historic N deposition maps | Yes | Yes | Yes |
| Historic N fertilizer application maps | No | Yes | Yes |
| Historic domestic grazing-ruminant density maps | No | No | Yes |
| Wild herbivores density maps | Yes | No | No |



Table 2. Grass-biomass production deficits in regions where simulated productivity by ORCHIDEE-GM Sim-GM (see text) cannot fulfil the grass-biomass use given by Herrero et al. (2013) for 2000.

| Regions[a] | Grass biomass use (million tonne DM) | Production deficit (million tonne DM) | $f_{deficit}$ (%)[b] | $f_{global}$ (%)[c] |
|---|---|---|---|---|
| North America | 228 | 24 | 11% | 6% |
| Russian Federation | 52 | 2 | 4% | 1% |
| Western Europe | 196 | 10 | 5% | 2% |
| Eastern Europe | 82 | 3 | 4% | 1% |
| Near East & North Africa | 175 | 67 | 38% | 16% |
| East & Southeast Asia | 275 | 34 | 12% | 8% |
| Oceania | 107 | 4 | 4% | 1% |
| South Asia | 390 | 209 | 53% | 50% |
| Latin America & Caribbean | 534 | 20 | 4% | 5% |
| Sub-Saharan Africa | 351 | 46 | 13% | 11% |
| World total | 2391 | 420 | 18% | 100% |

[a] Regions are classified following the definition in the FAO Global Livestock Environmental Assessment Model (GLEAM; http://www.fao.org/gleam/en/).

5    [b] $f_{deficit}$ is the fraction of production deficit in the total grass biomass use of the region for 2000.

[c] $f_{global}$ is the fraction of production deficit in the global total production deficit for 2000.



Table 3. Area and mean productivity of managed grassland from this study, ruminant numbers, and pasture area from HYDE 3.1 dataset for 1901 and 2000 by regions and global total.

| Regions[a] | Grassland area (1000 km$^2$; 1901/2000) | | | Mean Productivity (g DM m$^2$ yr$^{-1}$; 1900s/1990s[b]) | | $N_{ruminant}$ [c] (10$^6$ LU; 1901/2000) | Pasture area from HYDE 3.1[d] (1000 km$^2$; 1901/2000) |
|---|---|---|---|---|---|---|---|
| | Total managed | Mown | Grazed | $Y_{cut}$ | $Y_{graze}$ | | |
| North America | 925/1203 | 47/96 | 877/1108 | 280/399 | 114/156 | 42/87 | 1157/2482 |
| Russian Federation | 274/441 | 14/29 | 260/412 | 220/437 | 79/101 | 9/16 | 2995/904 |
| Western Europe | 480/517 | 41/31 | 439/486 | 505/840 | 225/316 | 49/76 | 793/595 |
| Eastern Europe | 317/346 | 52/44 | 265/302 | 282/550 | 118/197 | 12/17 | 655/248 |
| Near East & North Africa | 420/1062 | 15/139 | 405/923 | 108/193 | 60/79 | 12/50 | 2607/5607 |
| East & Southeast Asia | 362/1247 | 6/88 | 356/1159 | 403/727 | 106/161 | 14/83 | 2998/5327 |
| Oceania | 321/696 | 9/60 | 312/636 | 205/332 | 98/122 | 11/33 | 979/4000 |
| South Asia | 497/688 | 84/200 | 412/488 | 333/593 | 122/154 | 35/109 | 651/962 |
| Latin America & Caribbean | 842/2473 | 13/60 | 830/2413 | 294/346 | 134/204 | 40/194 | 1341/5446 |
| Sub-Saharan Africa | 635/2282 | 17/108 | 618/2174 | 249/466 | 91/111 | 16/93 | 4486/6991 |
| Global total | 5072/10955 | 297/855 | 4775/10100 | 312/484 | 117/156 | 238/759 | 19181/32764 |

[a] Regions are classified following the definition in the FAO Global Livestock Environmental Assessment Model (GLEAM; http://www.fao.org/gleam/en/).

[b] The potential harvested biomass from mown grassland ($Y_{cut}$) and the potential biomass consumption over grazed grassland ($Y_{graze}$) are 10-year averages for the period 1901-1910 (1900s) and 1991-2000 (1990s) representing the productivity at the beginning and at the end of the 20th century respectively.

[c] Ruminant numbers (in units of Livestock Unit, LU) are calculated based on the total metabolisable energy (ME) requirement by all ruminant. The ME requirement by all ruminants is based on ruminant numbers from statistics (for 1961-2021; data derived from FAOSTAT, 2013) and literature estimates (for 1901-1960; data derived from Mitchell (1993, 1998) and available in HYDE database at: http://themasites.pbl.nl/tridion/en/themasites/hyde/landusedata/livestock/index-2.html), using the calculation method given in the Supporting Information Text S1 of Chang et al. (2015b).

d see Klein Goldewijk et al. (2011) for details.



Table 4. Comparison between modelled GPPs and MODIS-GPP. The spatial pattern (*Slope* and $r_{spatial}$) and the interannual variability ($r_{IAV}$) are compared.

|  | $GPP_{Sim-GU}$ vs. MODIS-GPP | $GPP_{Sim-GM}$ vs. MODIS-GPP |
|---|---|---|
| *Slope* | 1.01 | 1.04 |
| $r_{spatial}$ | 0.85 | 0.84 |
| $r_{IAV-global}$ | 0.88 | 0.87 |
| $r_{IAV}$ | $0.39 \pm 0.33$ | $0.38 \pm 0.34$ |

*Slope* and $r_{spatial}$: slope and correlation coefficient from the linear regressions between modelled and MODIS-GPP for grid-cells with grassland covering more than 20% of total land in the MOD12Q1 dataset.

$r_{IAV-global}$: correlation coefficient between detrended time-series of global total modelled GPP and MODIS-GPP from 2000 to 2013.

$r_{IAV}$: correlation coefficient between detrended time-series of modelled GPP and MODIS-GPP at grid-cell level. Here, we show the average ± standard deviation of $r_{IAV}$ for all grid-cells with grassland covering more than 20% of total land in the MOD12Q1 dataset. The average ± standard deviation of $r_{IAV}$ spatial resolutions $0.5^o \times 0.5^o$ is presented.



Table 5. Mean ± standard deviation of $r_{seasonal}$ comparing the seasonal cycle of modelled GPP, MODIS-GPP and SIF data for the five latitude bands and global scale. $r_{seasonal}$ is expressed as mean ± standard deviation of grid level correlation coefficient within each latitude band and global. To avoid the strong impact of other land cover types (e.g., crop and forest) to the seasonal cycle, we only consider $r_{seasonal}$ for grid-cells with grassland covering more than 50% of total land in the MOD12Q1 dataset.

| $r_{seasonal}$ | Latitude bands | | | | | Global |
|---|---|---|---|---|---|---|
| | 60°N - 90°N | 30°N - 60°N | 0 - 30°N | 0 - 30°S | 30°S - 60°S | |
| $GPP_{Sim-GU}$ vs. SIF data | 0.84 ± 0.15 | 0.81 ± 0.19 | 0.65 ± 0.27 | 0.68 ± 0.28 | 0.56 ± 0.33 | 0.77 ± 0.23 |
| $GPP_{Sim-GM}$ vs. SIF data | 0.84 ± 0.15 | 0.81 ± 0.19 | 0.65 ± 0.27 | 0.68 ± 0.29 | 0.55 ± 0.33 | 0.77 ± 0.23 |
| $GPP_{Sim-GU}$ vs. MODIS-GPP | 0.89 ± 0.10 | 0.86 ± 0.16 | 0.70 ± 0.30 | 0.63 ± 0.43 | 0.65 ± 0.32 | 0.80 ± 0.27 |
| $GPP_{Sim-GM}$ vs. MODIS-GPP | 0.89 ± 0.10 | 0.86 ± 0.16 | 0.70 ± 0.30 | 0.62 ± 0.44 | 0.64 ± 0.32 | 0.80 ± 0.28 |
| MODIS-GPP vs. SIF data | 0.90 ± 0.11 | 0.87 ± 0.16 | 0.80 ± 0.22 | 0.61 ± 0.37 | 0.61 ± 0.36 | 0.81 ± 0.25 |





**Figures**

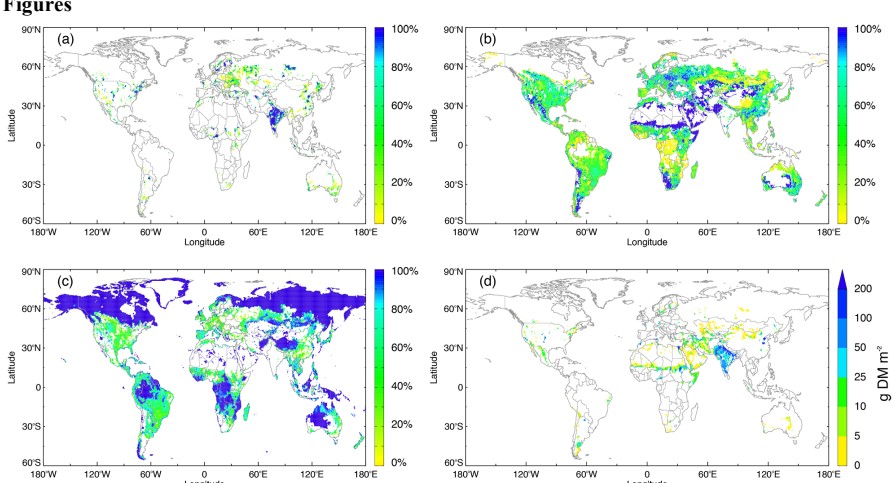

Figure 1. (a) Mown, (b) grazed, and (c) unmanaged fraction of global grassland, and (d) modelled grass-biomass production deficit of 2000. Modelled grass-biomass production deficit indicates the simulated grassland productivity in the grid cells is not sufficient to fulfill the grass-biomass use given by Herrero et al. (2013), and is expressed with units of g dry matter (DM) per $m^2$ of total land area in each grid cell.



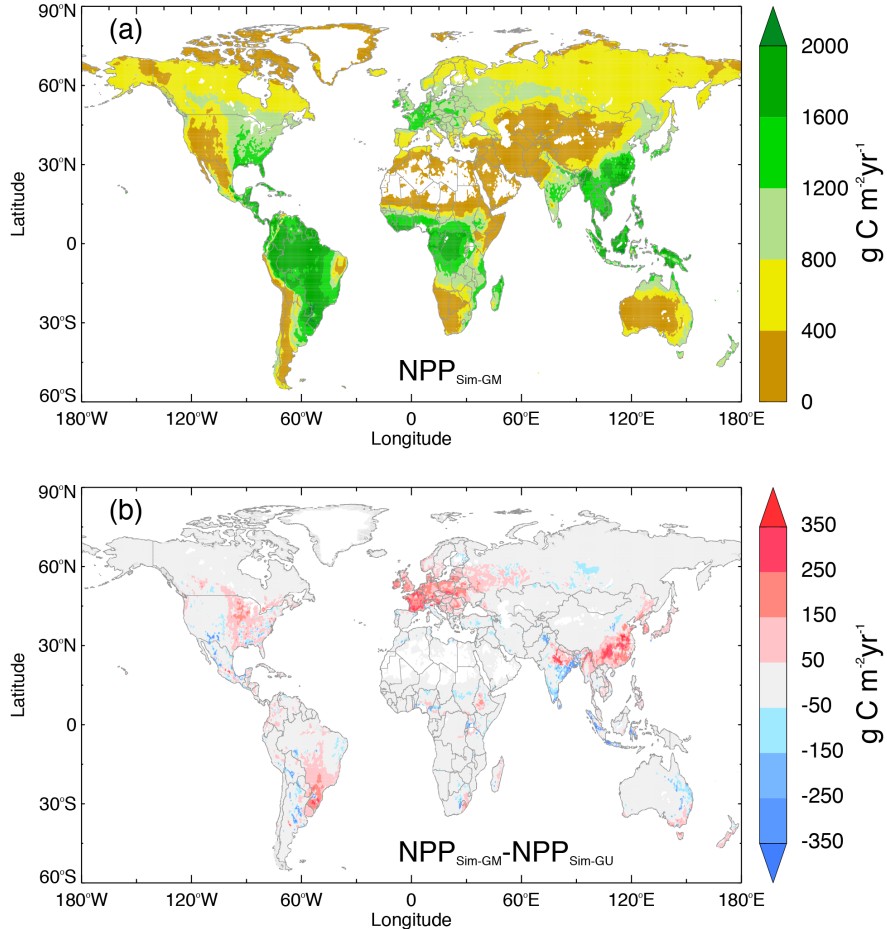

Figure 2. Modelled mean grassland NPP for the period 1990-1999 from the simulation experiments Sim-GM (a), and the NPP differences (b) between Sim-GM and Sim-GU. See Sect. 2.5 for the details of the experiments.




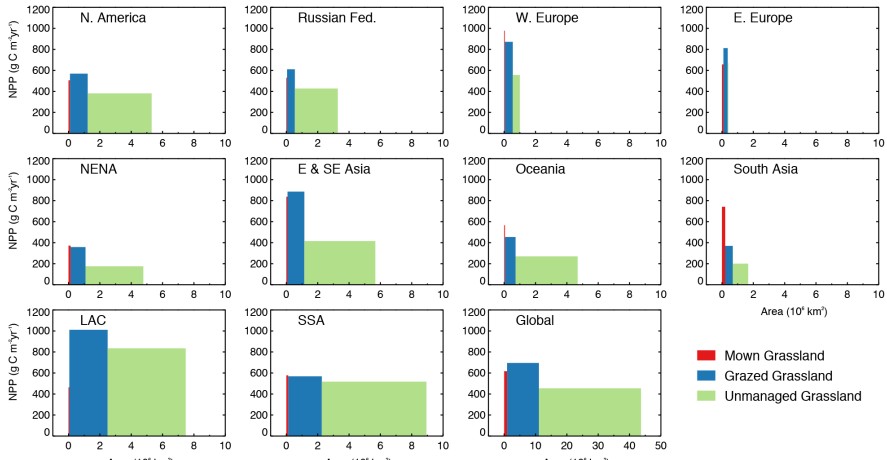

Figure 3. Productivities per unit area (height of each rectangle) and grassland areas (width of each rectangle) of the different types of grassland (mown, grazed, and unmanaged grassland) by FAO-defined regions and global total. Areas in the graph shows the production of each grassland type. Productivities and grassland areas are averaged for 1991-2000 from experiment Sim-GM. The FAO-defined regions (from top-left) are North America, Russian Federation, Western Europe, Eastern Europe, Near East & North Africa (NENA), East & Southeast Asia, Oceania, South Asia, Latin America and the Caribbean (LAC), Sub-Saharan Africa (SSA).





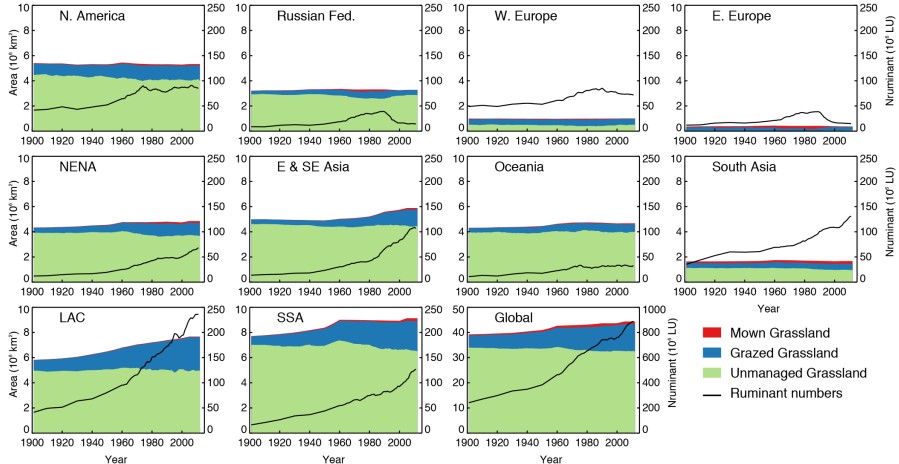

Figure 4. Historic changes in the area of managed/unmanaged grassland, and in the ruminant numbers for 1901 and 2012 by regions and global total. See caption to Fig. 3 for expansion of FAO-defined regions.





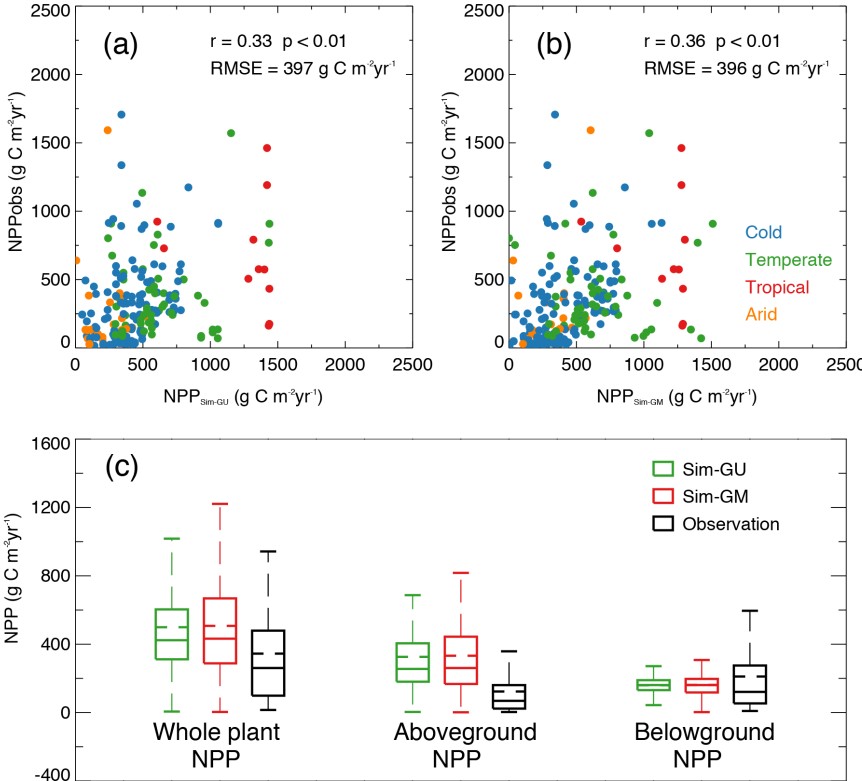

Figure 5. Comparison between site-observations of whole plant NPP ($NPP_{obs}$) and modelled NPP from experiments: (a) Sim-GU or (b) Sim-GM, and (c) box-and-whisker plot of the observed and modelled annual whole-plant NPP, aboveground NPP and belowground NPP. In subplot (a) and (b), grassland sites in different Köppen climate zones are specified by different colours. The Köppen climate zones are classified based on Peel et al. (2007) using climate data from WorldClim (http://www.worldclim.org/). In subplot (c), NPPs from different experiments are specified by different colours, and the 'whisker' indicates the cross-measurement (total 214 measurements) uncertainty.





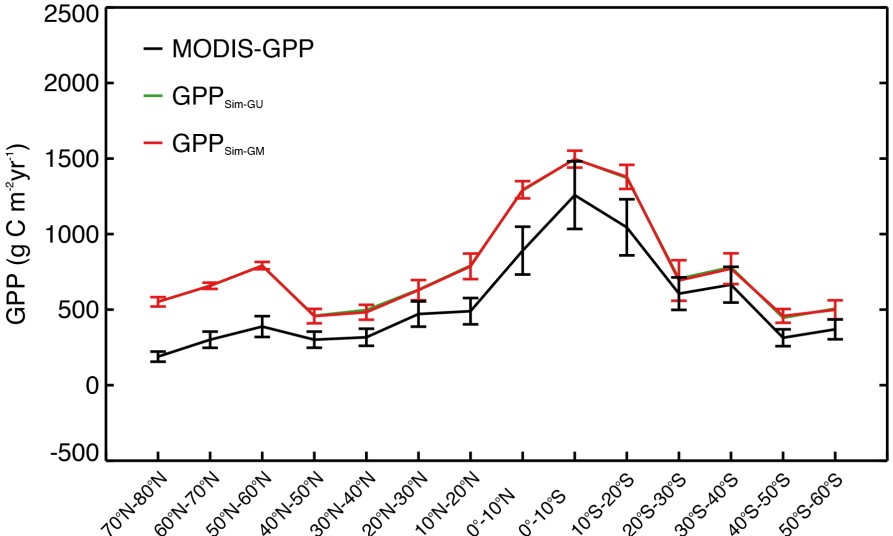

Figure 6. Comparison between MODIS-GPP and modelled GPP from two experiments, by latitude band. $GPP_{Sim-GU}$ (green line) is nearly fully covered by $GPP_{Sim-GM}$ (red line). The uncertainty of MODIS-GPP comes from the reported relative error term driven by NASA's Data Assimilation Office (DAO) reanalysis datasets (Zhao et al., 2006). The uncertainty of modelled GPP is the standard deviation of interannual variation of grassland GPP in each band for the period 2000-2013. To make the figure less complicated and readable, only the uncertainty of $GPP_{Sim-GM}$ is presented as an example.





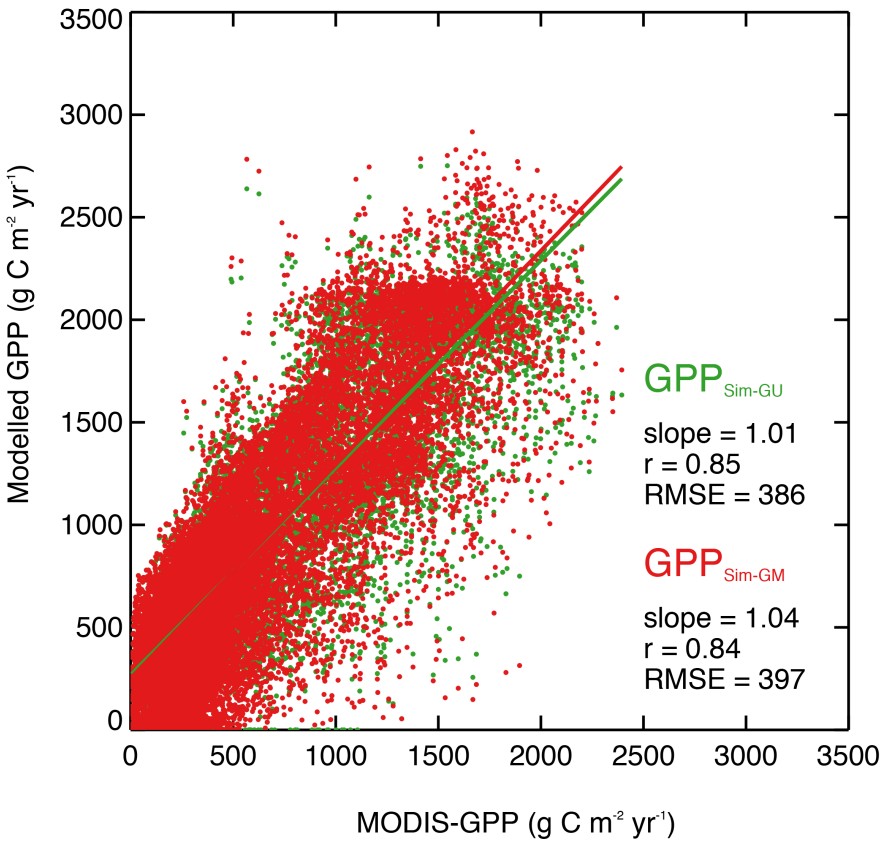

Figure 7. Comparison between MODIS-GPP and modelled GPP from two experiments at the resolution of $0.5^{\circ} \times 0.5^{\circ}$.





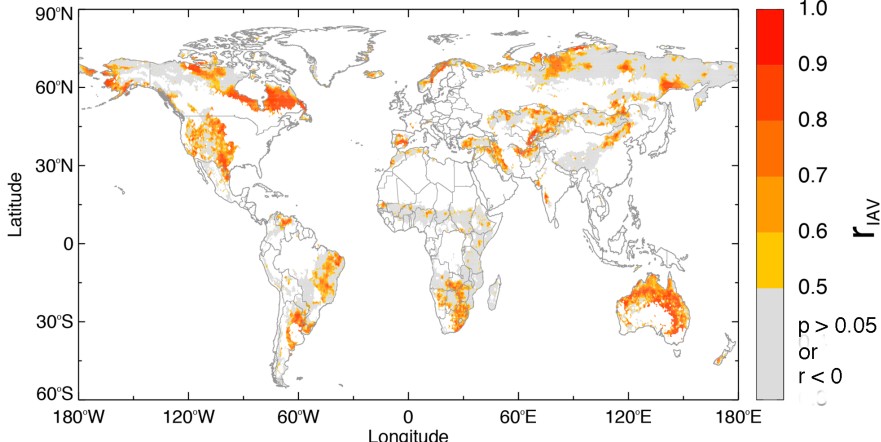

Figure 8. Spatial distribution of $r_{IAV}$ between MODIS-GPP and $GPP_{Sim-GM}$. $r_{IAV}$ is the correlation coefficient between detrended time-series of modelled and MODIS-GPP from 2000 to 2013. This figure only shows the $r_{IAV}$ for grid-cells with grassland covering more than 20% of total land in the MOD12Q1 dataset. Grey colour indicates insignificant or negative $r_{IAV}$ ($p > 0.05$ or $r < 0$); and yellow-to-red indicate significant positive $r_{IAV}$ with increasing value ($r > 0$ and $p < 0.05$).





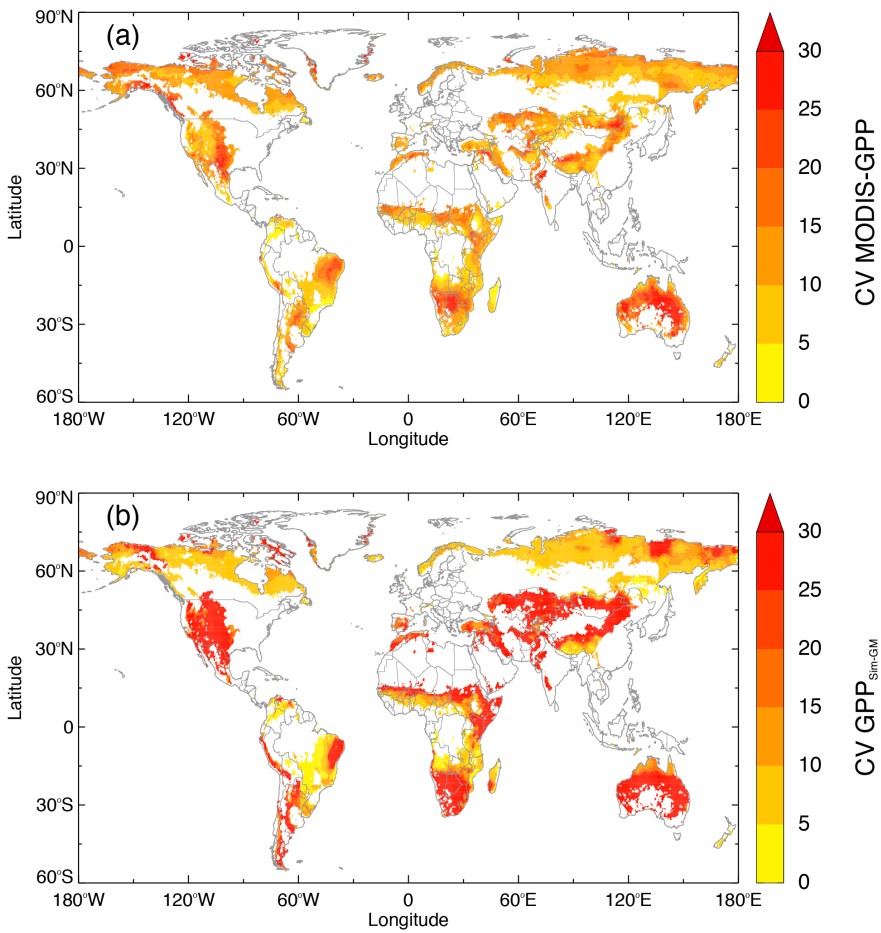

Figure 9. Coefficient of variation (CV) of (a) MODIS-GPP and (b) $GPP_{Sim\text{-}GM}$.




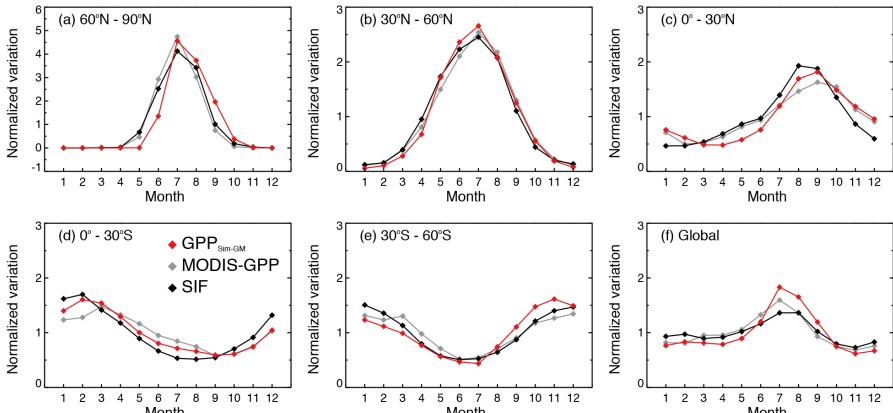

Figure 10. The normalized seasonal variation of modelled GPP ($GPP_{Sim-GM}$), MODIS-GPP, and SIF for five latitude bands (a – e) and (f) global average.



Figure 11. Spatial distribution of $r_{seasonal}$ between (a) SIF data and $GPP_{Sim-GM}$, (b) MODIS-GPP and $GPP_{Sim-GM}$, and (c) MODIS-GPP and SIF data respectively. $r_{seasonal}$ is the correlation coefficient





between mean seasonal cycle of modelled GPP, MODIS-GPP and SIF data from 2008 to 2012. This figure only shows the $r_{seasonal}$ for grid-cells with grassland covering more than 50% of total land in the MOD12Q1 dataset. Grey colour indicates insignificant or negative $r_{seasonal}$ ($p > 0.05$ or $r < 0$); and yellow-to-red indicate significant positive $r_{seasonal}$ with increasing value ($r > 0$ and $p < 0.05$).





Figure 12. The maximum monthly GPP ($GPP_{max}$) from (a) ORCHIDEE-GM v3.1 ($GPP_{Sim-GM}$), (b) MODIS-GPP and (c) GPP derived from SIF Version 26 (SIF-GPP). Data are monthly average for the period 2008 - 2012.





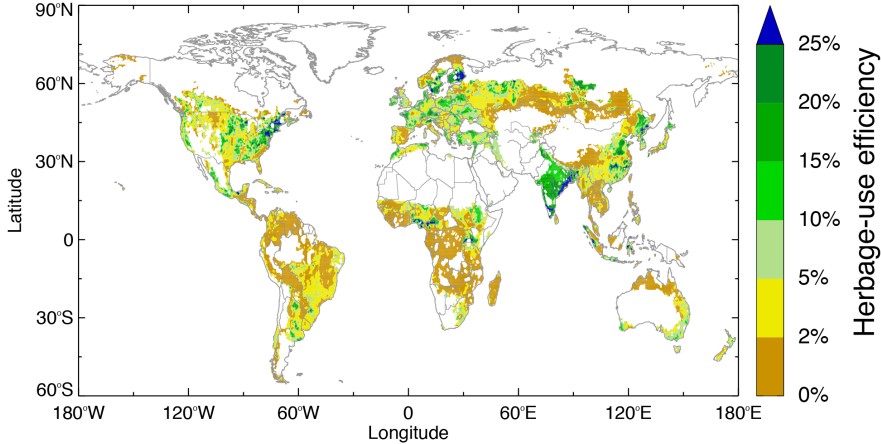

Figure 13. Average herbage-use efficiency over managed grassland (grazed plus mown) in 2000-2009 simulated by ORCHIDEE-GM v3.1. Herbage use efficiency (Hodgson, 1979) is defined as the forage removed expressed as a proportion of herbage growth. In this study, the forage removed is modelled annual grass biomass use including $Y_{grazed}$ and $Y_{mown}$, and herbage growth is modeled annual grass GPP.