# Peer review of "Combining livestock production information in a process based vegetation model to reconstruct the history of grassland management"

_Biogeosciences, 2016_

## Referee Comment (RC1) · Anonymous Referee #1 · 31 Mar 2016

The manuscript estimates globally the historical management intensity of grasslands. Thereby, authors use the process-based vegetation model ORCHIDEE-GM in combination with globally derived maps on livestock density, wild herbivory density, nitrogen fertilization and atmospheric nitrogen deposition, and grass-biomass use. Authors can show that largest fractions of managed grasslands occur in regions of high livestock density. A comparison of grassland productivity between managed and unmanaged grassland simulations shows that management has largest impact in regions of high N fertilizer applications. Authors further examined a global increase of 116% of managed grassland area (from 5.1x106 km2 in 1901 to 11x106 km2 in 2000). The topic is interesting and scientifically relevant as more research focusses on the global impact of

land use but historical data on land use is rare. Nevertheless, the manuscript requires large improvements.

I miss a clear statement on the hypothesis or goal of this study in the introduction. While reading the manuscript, it was confusing if authors focus on global management intensity, net biome productivity (NBP) or grassland productivity (NPP). Previous studies and intentions of the study presented in this manuscript are mixed so that it is confusing which parts of this study are novel and which parts are used from previous studies. Is the presented study just an extension of the Europe-study of Chang et al. 2015a? Which challenges arise by constructing a management intensity map for the globe instead of only Europe? Are there differences in the methodology? I highly recommend (1.) providing a clear statement on the goal of this study, (2.) highlighting challenges which arise and (3.) indicating the authors' own novel contribution for achieving this goal. The results and discussion section should also be more focused, following the hypotheses or goals that should be formulated clearly in the introduction.

Besides the motivation of this study, the methods section requires large clarification in a similar way. For the model description the authors write about applications of recent model versions (v1 and v2.1) and state that they use version 3.1 of ORCHIDEE-GM. However, I would expect (especially for readers who are not familiar with ORCHIDEE and ORCHIDEE-GM) to get basic information on the model (i.e. most important modelled processes, time step, spatial scale, important input and output of the model). Concerning the model parameters in section 2.2, only 2 parameters are mentioned. Information on where to find the other parameters of the model and their values should be provided. Moreover, this paragraph occurs a second time in the supplement (which is just redundant information). The text S1 in the supplement is, however, written much better and more concise than in the main manuscript. This applies also for the other text paragraphs in the manuscript of section 2.3 and their corresponding text in the supplement. Partly, introductory information occurs in the supplementary paragraph while it is needed in the paragraph of the main manuscript. In turn, technical information

occurs in the main manuscript which is hard to understand without reading the supplementary text first. Following sections 2.4 and 2.5, it's difficult to understand which maps provide input for ORCHIDEE-GM simulations and which maps are combined with simulation output of ORCHIDEE-GM. In total, the entire methods section needs large improvements, i.e. clear, concise and comprehensive statements in order to be able to reproduce the results of this study.

Regarding the manuscript language and style, I highly recommend to shorten the manuscript and to be more concise and precise, but still comprehensive. The entire manuscript is too long. Sentences are too long to fluently read the manuscript, some paragraphs are too technical. There are grammar and spelling mistakes. References should be double-checked (e.g., page 4, line 12). The last sentences of the abstract (page 2, lines 13-21) are confusingly written and hard to understand without reading the entire article.

---

## Referee Comment (RC2) · Anonymous Referee #2 · 5 Apr 2016

General comments: This study attempted to reconstruct the history of grassland management by integrating grazing-ruminant stocking density maps, wild-herbivores population density maps, nitrogen fertilizer application maps as well as nitrogen deposition maps to develop grassland management intensity maps. This land use information is very important to global change studies and very interesting as well. The attempt of integrating those scattered data in various scales is valuable even though the methods might be over-simplified. The manuscript, however, poorly delivered this information. I think the title of this manuscript delivered interesting and clear information about the study, but the main text lost focus that were specified in the title and the abstract. The method sections (in both the main text and the SI) are very confusing and could be

more organized. Some descriptions on modeling or calculation were unnecessarily complicated, and some assumptions for extrapolating data need to be checked carefully. Overall, the current version requires major revisions before considered for publication.

Specific comments: [ORCHIDEE-GM model] (1) 'Results' and 'Discussion' of the current version made this manuscript read like evaluating the performance of the updated version of ORCHIDEE-GM model that includes livestock data to estimate global grass biomass. The model is a key piece in this study, which generates the NPP and GPP, but it seems the goal of this study is actually 'combining livestock production' and 'to reconstruct the history of grassland management'. If so, the main text should be reorganized. The evaluation-related sections could be combined. (2) The model-related descriptions in the 'Material and Methods' section are not clear. At page 4 line 28-32, it is not clear what was updated in the model v3.1. Only bug-corrections? Are there any updates in modeling ecological processes or management activities? (3) At page 5 line 22-25, the author listed the input data, but the output was never clearly described in the manuscript. This information may be described in previous publications, but it would be good to briefly describe in this manuscript. Line 12-15 at page 7 reads like descriptions of output, but confusing. I think this part is very important as it is related to how the authors defined and quantified 'management intensity', so it needs to be clearly presented. (4) Does '... not use a land-cover map in the simulations, but rather consider that grasslands are distributed all over the world' mean the areas that are not characterized as grassland in a land-cover map have zero grass productivities in your productivity maps? (5) Line 14-15 at page 8, how the $Y_{grazed}$ is calculated from $D_{grazing,m,k}$? I think this is a key step of this study and should be described clearly. [Variables, equations and data conversions] There are many equations and data conversions in this manuscript. The authors should define variables clearly and present units for important variables (e.g. D in text S2), so that the readers can easily follow the ideas of producing those data sets. Or, a table listing those variables and associated data sources might be helpful. (6) I think the

assumption at Line 4-5 at page SI_3 might be wrong as the ratio of the total ruminant density between years can be calculated based on the assumptions in text S2. I could be wrong, but I think the authors should carefully check the conversion and should not make too many assumptions arbitrarily as this might affect the results significantly. A brief interpretation of my thoughts: see the supplement for equations and calculations. (7) This point may be trivial, so it is just a suggestion. I don't think the variable of ME index ($I_{m,j}$, page 8 and page SI_3) is really necessary unless the ME index has some other meanings. The assumptions seemed just to be: see the supplement. The ME index made the conversions more complicated than it should be.

Please also note the supplement to this comment:
http://www.biogeosciences-discuss.net/bg-2016-8/bg-2016-8-RC2-supplement.pdf

**Supplement:**

**Combining livestock production information in a process based vegetation model to reconstruct the history of grassland management**

**Journal: Biogeosciences Discuss.**

**General comments:**

This study attempted to reconstruct the history of grassland management by integrating grazing-ruminant stocking density maps, wild-herbivores population density maps, nitrogen fertilizer application maps as well as nitrogen deposition maps to develop grassland management intensity maps. This land use information is very important to global change studies and very interesting as well. The attempt of integrating those scattered data in various scales is valuable even though the methods might be over-simplified. The manuscript, however, poorly delivered this information. I think the title of this manuscript delivered interesting and clear information about the study, but the main text lost focus that were specified in the title and the abstract. The method sections (in both the main text and the SI) are very confusing and could be more organized. Some descriptions on modeling or calculation were unnecessarily complicated, and some assumptions for extrapolating data need to be checked carefully. Overall, the current version requires major revisions before considered for publication.

**Specific comments:**

ORCHIDEE-GM model

(1) 'Results' and 'Discussion' of the current version made this manuscript read like evaluating the performance of the updated version of ORCHIDEE-GM model that includes livestock data to estimate global grass biomass. The model is a key piece in this study, which generates the NPP and GPP, but it seems the goal of this study is actually 'combining livestock production' and 'to reconstruct the history of grassland management'. If so, the main text should be reorganized. The evaluation-related sections could be combined.

(2) The model-related descriptions in the 'Material and Methods' section are not clear. At page 4 line 28-32, it is not clear what was updated in the model v3.1. Only bug-corrections? Are there any updates in modeling ecological processes or management activities?

(3) At page 5 line 22-25, the author listed the input data, but the output was never clearly described in the manuscript. This information may be described in previous publications, but it would be good to briefly describe in this manuscript. Line 12-15 at page 7 reads like descriptions of output, but confusing. I think this part is very important as it is related to how the authors defined and quantified 'management intensity', so it needs to be clearly presented.

(4) Does '… not use a land-cover map in the simulations, but rather consider that grasslands are distributed all over the world' mean the areas that are not characterized as grassland in a land-cover map have zero grass productivities in your productivity maps?

(5) Line 14-15 at page 8, how the $Y_{grazed}$ is calculated from $D_{grazing,m,k}$? I think this is a key step of this study and should be described clearly.

Variables, equations and data conversions

There are many equations and data conversions in this manuscript. The authors should define variables clearly and present units for important variables (e.g. D in text S2), so that the readers can easily follow the ideas of producing those data sets. Or, a table listing those variables and associated data sources might be helpful.

(6) I think the assumption at Line 4-5 at page SI_3 might be wrong as the ratio of the total ruminant density between years can be calculated based on the assumptions in text S2. I could be wrong, but I think the authors should carefully check the conversion and should not make too many assumptions arbitrarily as this might affect the results significantly. A brief interpretation of my thoughts:

From Equations S1, S2 and S3:

$$D_{m,k} = \left( \sum_i D_{i,k} \times F_{i,j} \right)_m = \left( \sum_i \frac{N_{i,k}}{A_k} \times \frac{ME_{i,j}}{N_{i,j} \times ME_{LU}} \right)_m$$

$$= \frac{1}{A_k ME_{LU}} \left( \sum_i \frac{N_{i,k}}{N_{i,j}} \times ME_{i,j} \right)_m = \frac{1}{A_k ME_{LU}} \left( \sum_i r_{i,k} \times ME_{i,j} \right)_m$$

$$\frac{D_{m1,k}}{D_{m2,k}} = \frac{\left( \sum_i r_{i,k} \times ME_{i,j} \right)_{m1}}{\left( \sum_i r_{i,k} \times ME_{i,j} \right)_{m2}}$$

in which $A_k$ is the area of grid $k$ and $r_{i,k}$ is the ratio of the number of livestock category $i$ in grid $k$ to the one in country $j$. This number can be easily calculated based on GLW v 2.0 dataset, and I believe it is not constant for different livestock categories and in all grids. But:

$$\frac{ME_{m1,j}}{ME_{m2,j}} = \frac{\left( \sum_i ME_{i,j} \right)_{m1}}{\left( \sum_i ME_{i,j} \right)_{m2}}$$ . If assuming $\frac{D_{m1,k}}{D_{m2,k}} = \frac{ME_{m1,j}}{ME_{m2,j}}$, that means $r_{i,k}$ is assumed constant for all animals and across all grids in country $j$, which could be wrong if the GLW v 2.0 dataset shows it is not.

(7) This point may be trivial, so it is just a suggestion. I don't think the variable of ME index ($I_{m,j}$, page 8 and page SI_3) is really necessary unless the ME index has some other meanings. The assumptions seemed just to be: $\frac{GBU_{m1,k}}{GBU_{m2,k}} = \frac{ME_{m1,j}}{ME_{m2,j}}$ and $\frac{D_{m1,k}}{D_{m2,k}} = \frac{ME_{m1,j}}{ME_{m2,j}}$. The ME index made the conversions more complicated than it should be.

---

## Author Comment (AC1) · 11 May 2016

The manuscript estimates globally the historical management intensity of grasslands. Thereby, authors use the process-based vegetation model ORCHIDEE-GM in combination with globally derived maps on livestock density, wild herbivory density, nitrogen fertilization and atmospheric nitrogen deposition, and grass-biomass use. Authors can show that largest fractions of managed grasslands occur in regions of high livestock density. A comparison of grassland productivity between managed and unmanaged grassland simulations shows that management has largest impact in regions of high N fertilizer applications. Authors further examined a global increase of 116% of managed grassland area (from 5.1x106 km2 in 1901 to 11x106 km2 in 2000). The topic is interesting and scientifically relevant as more research focusses on the global impact of land use but historical data on land use is rare. Nevertheless, the manuscript requires large improvements.

**[Response]** We thank the reviewer for the valuable comments. Please find our detailed responses below each comment in blue. The corresponding major modifications in the revised manuscript were attached as Appendix A1-A4, Figure A1 and Table A1.

**[Comment 1]** I miss a clear statement on the hypothesis or goal of this study in the introduction. While reading the manuscript, it was confusing if authors focus on global management intensity, net biome productivity (NBP) or grassland productivity (NPP). Previous studies and intentions of the study presented in this manuscript are mixed so that it is confusing which parts of this study are novel and which parts are used from previous studies. Is the presented study just an extension of the Europe-study of Chang et al. 2015a? Which challenges arise by constructing a management intensity map for the globe instead of only Europe? Are there differences in the methodology? I highly recommend (1.) providing a clear statement on the goal of this study, (2.) highlighting challenges which arise and (3.) indicating the authors' own novel contribution for achieving this goal. The results and discussion section should also be more focused, following the hypotheses or goals that should be formulated clearly in the introduction.

**[Response]** Thanks for the suggestion. We have rephrased the 'Introduction' section as Appendix A1. In the revised introduction, we presented the importance of grassland management intensity history (paragraph 1 and 2), pointed out the limitations of the previous studies related to grassland management and the lack of the gridded management intensity history maps (paragraph 2). Then we cited a recent study that provides a starting point to the reconstruction in this study (paragraph 3). In the last paragraph of introduction, we presented the goal, and the structure of this study.

This study is beyond an extension of the Europe-study. We pointed out the limitation of previous study (paragraph 2) to emphasize the necessary of gridded information on management intensity and the long-term history (1901-2012), which does not exist before and is the challenge and novelty of this study.

In the revised manuscript, we reorganized the structure to better focus on the major goal of this study as reconstructing the history of grassland management intensity. Given the fact that the gridded grassland management intensity maps are productivity-dependent, we still give a specific attention to the evaluation of modeled productivity against both a new set of site-level NPP measurements, and satellite-based models of NPP and Gross Primary Productivity (GPP). The evaluation part has been combined and shortened in the revised manuscript.

**[Comment 2]** Besides the motivation of this study, the methods section requires large clarification in a similar way. For the model description the authors write about applications of recent model versions (v1 and v2.1) and state that they use version 3.1 of ORCHIDEE-GM. However, I would expect (especially for readers who are not familiar with ORCHIDEE and ORCHIDEE-GM) to get basic information on the model (i.e. most important modelled processes, time step, spatial scale, important input and output of the model).

**[Response]** In section 2.1 of the revised manuscript, we have added some more basic information of ORCHIDEE and ORCHIDEE-GM (as Appendix A2) including 1) the major processes and output of ORCHIDEE and their time step, and 2) the processes and output of the management module and the time step. The spatial scale is presented in the previous manuscript as "from site-level to global scale". The important input of the model in this study was presented in the section 2.4 'Model input' of the revised manuscript.

**[Comment 3]** Concerning the model parameters in section 2.2, only 2 parameters are mentioned. Information on where to find the other parameters of the model and their values should be provided. Moreover, this paragraph occurs a second time in the supplement (which is just redundant information). The text S1 in the supplement is, however, written much better and more concise than in the main manuscript.

**[Response]** Original section 2.2 in the previous manuscript and Text S1 has been combined as the section 2.2 in the revised manuscript. In addition, the reference on the other model parameterization was added as "All other parameters of ORCHIDEE model are kept consistent with that in Trunk.rev2425. The parameter settings for grassland management module are in consistent with that in ORCHIDEE-GM v1 (Chang et al., 2013) and v2.1 (Chang et al., 2015ab)".

**[Comment 4]** This applies also for the other text paragraphs in the manuscript of section 2.3 and their corresponding text in the supplement. Partly, introductory information occurs in the supplementary paragraph while it is needed in the paragraph of the main manuscript. In turn, technical information occurs in the main manuscript which is hard to understand without reading the supplementary text first.

**[Response]** Original section 2.3 in the previous manuscript has been separated to 2 sections: 2.4 Model input; and 2.5 Simulation set-up. The paragraphs have been rephrased with introductory information and only necessary technical information (as Appendix A3), and the corresponding text in the supplementary information is reorganized and rephrased too.

**[Comment 5]** Following sections 2.4 and 2.5, it's difficult to understand which maps provide input for ORCHIDEE-GM simulations and which maps are combined with simulation output of ORCHIDEE-

GM. In total, the entire methods section needs large improvements, i.e. clear, concise and comprehensive statements in order to be able to reproduce the results of this study.

[Response] In the revised manuscript, we have added a new flowchart (Fig. A1) illustrating the procedures for reconstructing the management intensity history, and a table listing all variables shown in the method section (including abbreviation, units, related equations, and data sources). We believe that the flowchart and the revised section 2.4-2.6 presented the reconstruction of the grassland management intensity maps in a more comprehensive way than before.

[Comment 6] Regarding the manuscript language and style, I highly recommend to shorten the manuscript and to be more concise and precise, but still comprehensive. The entire manuscript is too long. Sentences are too long to fluently read the manuscript, some paragraphs are too technical. There are grammar and spelling mistakes. References should be double-checked (e.g., page 4, line 12).

[Response] In the revised manuscript, we have reorganized the manuscript through 1) rephrasing section 2.4 'Model input' with only introductory information and necessary technical information; 2) combining the previous section 2.5 'Modelled productivity', 2.6 'Datasets for model evaluation' and section 2.7 'Model-data agreement matrics' as section 2.7 'Model evaluation' in the revised manuscript, 3) combining the model evaluation sections (section 3.2, 3.4 – 3.6 in the previous manuscript), and 4) shortening the discussion on productivity evaluation (section 4.3). However, we were not able to significantly reduce the size of this manuscript, because the comprehensive explanations of the critical material, key methods and results are necessary to help readers understanding the reconstruction of management intensity history in this study.

Thanks for the suggestions. We have corrected the grammar and spelling mistake and double-checked the reference in the revised manuscript. For example, the reference for PaSim model has been corrected as Riedo et al., 1998; Vuichard et al., 2007a,b; Graux et al., 2011. We have shortened or separated some long sentences to present them more clearly.

[Comment 7] The last sentences of the abstract (page 2, lines 13-21) are confusingly written and hard to understand without reading the entire article.

[Response] Given the reason that "the gridded grassland management intensity maps are model-dependent because they depend on modelled productivity", we gave a specific attention to the evaluation of modelled productivity in this study. We have deleted some detail information, and rephrased the last sentences of the abstract as Appendix A4.

**Appendix A1: Revised introduction**

[revised manuscript text omitted]

**Grazing-ruminant stocking density maps**. Spatial statistical information on grazing-ruminant stocking density (i.e., stocking rates) is not available at global scale. In this study, assuming that all the

ruminants in each grid-cell were grazing on the grassland within the same grid, we defined the grazing-ruminant stocking density in grid-cell $k$ in year $m$ ($D_{grazing,m,k}$, unit: LU per ha of grassland area) as:

$$D_{grazing,m,k} = \frac{D_{m,k}}{f_{grass,m,k}}$$
(1)

where $D_{m,k}$ is the total domestic ruminant stocking density in unit of LU per hectare of land area (Supplementary Information Text S1); and $f_{grass,m,k}$ is the grassland fraction in grid-cell $k$ in year $m$ from a set of historic land-cover change maps (Supplementary Information Text S2). To avoid unrealistic densities of ruminant grazing over grassland (which might cause grasses die during the growing season), a maximum value of 5 LU ha$^{-1}$ was set for the density map. In addition, a minimum grazing-ruminant density of 0.2 LU ha$^{-1}$ was set to avoid economically implausible stocking rates. The domestic ruminant stocking density (D, unit: LU per ha of land area) for the reference year 2006 is derived from the Gridded Livestock of the World v2.0 dataset (GLW v2.0; Robinson et al., 2014). The original density of each ruminant category (including cattle, sheep and goats, unit: head) is converted to Livestock Units (LU) and aggregated to the resolution of $0.5^{o} \times 0.5^{o}$. The category-specific gridded ruminant stocking density is then back-casted from 2012 to 1901 assuming that it has changed in each grid-cell proportionally with country-scale metabolisable energy requirement (ME) from that category of ruminants (Supplementary Information Text S1). The evolution of ME requirement by each category of ruminants was calculated from FAO ruminant population statistics during the period 1961-2012 (FAOSTAT, 2013) and from Mitchell (1993, 1998a, b) during the period 1901-1960 (http://themasites.pbl.nl/tridion/en/themasites/hyde/landusedata/livestock/index-2.html) using the method given in the Supporting information Text S1 of Chang et al. (2015b). Figure S1 shows the example maps of ruminant stocking density (D) and corresponding grazing-ruminant stocking density ($D_{grazing}$) for reference year 2006.

**Wild herbivore density maps.** Gridded maps of wild herbivore density are not available, therefore the gridded population density of wild herbivores ($D_{wild}$, unit: LU per ha of grassland area) is derived from the literature data, and from Bouwman et al. (1997) (see Table S2 for detail). The population of these herbivores was first converted to LU according to the ME requirement calculated from their mean weight (Table S2), and then distributed to suitable grasslands based on grassland aboveground (consumable) NPP simulated from ORCHIDEE-GM v3.1 (Supplementary Information Text S3; Fig. S2). The wild herbivores density was assumed to remain constant during the period of 1901-2012, because no gridded worldwide wild-animals population information was available.

**Nitrogen application rates from mineral fertilizers and manure.** Grassland is fertilized with organic nitrogen (N) fertilizer (e.g., manure, slurry) and/or even mineral-N fertilizer, though this is not as common as it is for cropland. Gridded fertilizer application rates on grassland are not available worldwide. The only exception that we are aware of is for European grasslands. Gridded mineral fertilizer and manure nitrogen application rates for grasslands for EU-27 was estimated by the CAPRI model (Leip et al., 2011, 2014) based on information from official and harmonized data sources such

as Eurostat, FAOstat and OECD, which are spatially disaggregated using the methodology described by Leip et al. (2008). For countries/region other than EU-27, the following data and methods were used (see Supplementary Information Text S4 for detail).

The amount of manure-N fertilizer for 17 world regions at 1995 was derived from various sources (e.g., IFA, 1999; FAO/IFA/IFDC, 1999; FAO/IFA, 2001) and synthesized by Bouwman et al. (2002a, b; Table S3). The regional data were downscaled to a 0.5° × 0.5° grid according to ruminant stocking density ($D$) of each grid-cell, which implies that locally higher ruminant density produces more manure. In each grid-cell, historical changes of manure-N fertilization ($N_{manure}$, unit: kg N per ha of grassland area per year) were assumed to follow the same evolution as the gridded total ruminant stocking density (including cattle, sheep and goats; Supplementary Information Text S1).

For mineral-N fertilizers on grassland, country-scale data of fertilized area and mean fertilization rate for 1999/2000 are available in FAO/IFA/IFDC/IPI/PPI (2002) with grassland/pasture been fertilized in 34 countries. Within the 34 countries, 21 of them belong to EU-27 where gridded fertilizer application rate is available. For the other 13 non-EU-27 countries, the national mean application rates (Table S4) are applied on grid-cells with a total ruminant stocking density above a certain threshold. The value of this threshold is determined for each country making the total grassland area of fertilized grids is identical to the national fertilized grassland area reported by FAO/IFA/IFDC/IPI/PPI (2002). The application rate of mineral-N fertilizers ($N_{mineral}$, unit: kg N per ha of grassland area per year) is extrapolated using country-scale total nitrogenous mineral fertilizers consumption data from FAOSTAT (2014) during the period 1961-2002. The mineral-N fertilization rate after 2002 is assumed to be constant as the 2002 rate. For the period 1901-1960, the same set of rules that were applied for the EU-27 (see section 'Simulation set-up' in Chang et al., 2015a for details) is used, namely: 1) no mineral-N fertilizer is applied over grasslands before 1950, and 2) for the period of 1951-1961, the rate of application is assumed to increase linearly from zero to the level of 1961 in each grid-cell.

[revised manuscript text omitted]

---

## Author Comment (AC2) · 11 May 2016

General comments: This study attempted to reconstruct the history of grassland management by integrating grazing-ruminant stocking density maps, wild-herbivores population density maps, nitrogen fertilizer application maps as well as nitrogen deposition maps to develop grassland management intensity maps. This land use information is very important to global change studies and very interesting as well. The attempt of integrating those scattered data in various scales is valuable even though the methods might be over-simplified. The manuscript, however, poorly delivered this information. I think the title of this manuscript delivered interesting and clear information about the study, but the main text lost focus that were specified in the title and the abstract. The method sections (in both the main text and the SI) are very confusing and could be more organized. Some descriptions on modeling or calculation were unnecessarily complicated, and some assumptions for extrapolating data need to be checked carefully. Overall, the current version requires major revisions before considered for publication.

**[Response]** We thank the reviewer for the valuable comments. Please find our detailed responses below each comment in blue. The corresponding major modifications in the revised manuscript were attached as Appendix A1-A3, Figure A1 and Table A1.

**[Comment 1]** (1) 'Results' and 'Discussion' of the current version made this manuscript read like evaluating the performance of the updated version of ORCHIDEE-GM model that includes livestock data to estimate global grass biomass. The model is a key piece in this study, which generates the NPP and GPP, but it seems the goal of this study is actually 'combining livestock production' and 'to reconstruct the history of grassland management'. If so, the main text should be reorganized. The evaluation-related sections could be combined.

**[Response]** Thanks for the suggestions. In order to stick to the goal of this study, we have revised the manuscript through 1) reorganizing section 2.4 - 2.6 to present the procedures of reconstructing grassland management intensity maps more clearly; 2) combining method sections on model evaluation to one subsection 2.7; 3) the model evaluation sections (section 3.2, 3.4 – 3.6 in the previous manuscript); and 4) shortening the discussion on model evaluation (section 4.3).

**[Comment 2]** (2) The model-related descriptions in the 'Material and Methods' section are not clear. At page 4 line 28-32, it is not clear what was updated in the model v3.1. Only bug-corrections? Are there any updates in modeling ecological processes or management activities?

**[Response]** In the version 3.1 of ORCHIDEE-GM, we made the adjustment of its parameters for the C4 grassland biome (Sect. 2.2), and implemented a specific strategy for wild animal grazing (Sect. 2.3). Furthermore, in the revised manuscript, version 3.1 has been updated with ORCHIDEE Trunk.rev2425 (a recent version of ORCHIDEE). The above information has been added in the section 2.1 of the revised manuscript.

**[Comment 3]** (3) At page 5 line 22-25, the author listed the input data, but the output was never clearly described in the manuscript. This information may be described in previous publications, but it would be good to briefly describe in this manuscript. Line 12-15 at page 7 reads like descriptions of output, but confusing. I think this part is very important as it is related to how the authors defined and quantified 'management intensity', so it needs to be clearly presented.

**[Response]** We have reorganized the sections in the revised manuscript to clarify the model input (section 2.4), simulation set-up (section 2.5), and the procedures for reconstructing management intensity history (section 2.6). Moreover, we have added a new flowchart (Fig. A1) illustrating the procedures for reconstructing management intensity history, and a table listing all variables shown in the method section (including abbreviation, units, related equations, and data sources). We believe that the flowchart and the revised section 2.4-2.6 presented the reconstruction of the grassland management intensity maps in a more comprehensive way than before.

**[Comment 4]** (4) Does '. . . not use a land-cover map in the simulations, but rather consider that grasslands are distributed all over the world' mean the areas that are not characterized as grassland in a land-cover map have zero grass productivities in your productivity maps?

**[Response]** During post-processing, the grids with zero grassland in the land-cover maps ($A_{grass,m,k} = 0$) will be masked, thus will have zero grass productivities in the productivity maps as shown in Fig. 2 in the previous manuscript. This clarification has been added in the section 2.5 'Simulation set-up' of the revised manuscript.

**[Comment 5]** (5) Line 14-15 at page 8, how the Ygrazed is calculated from Dgrazing,m,k? I think this is a key step of this study and should be described clearly. [Variables, equations and data conversions] There are many equations and data conversions in this manuscript. The authors should define variables clearly and present units for important variables (e.g. D in text S2), so that the readers can easily follow the ideas of producing those data sets. Or, a table listing those variables and associated data sources might be helpful.

**[Response]** Thanks for your suggestion. We added the description about how the model calculates the $Y_{grazed}$ and $Y_{mown}$ in the revised manuscript to clarify this key step (as Appendix A1). We also added a new table listing all variables shown in method section, including abbreviation, units, related equations, and data sources (Table A1).

**[Comment 6]** (6) I think the assumption at Line 4-5 at page SI_3 might be wrong as the ratio of the total ruminant density between years can be calculated based on the assumptions in text S2. I could be wrong, but I think the authors should carefully check the conversion and should not make too many assumptions arbitrarily as this might affect the results significantly. A brief interpretation of my thoughts: see the supplement for equations and calculations.

**[Response]** Thank you for the comment. Yes, you are right about the calculation. We should calculate the gridded ruminant density ($D_{m,k}$) variation and gridded grass biomass use ($GBU_{m,k}$) based on the category-specific variation of metabolisable energy (ME) requirement in the country rather than the changes in country-scale total ME requirement. Thus we have modified all related calculations

(including $D_{m,k}$, $D_{grazing,m,k}$, and $GBU_{m,k}$), re-run all simulations, and re-calculate gridded management intensity history based on modified calculation. In the revised manuscript, the calculations of $D_{m,k}$ and $GBU_{m,k}$ have been changed as Appendix A2 and Appendix A3 respectively. The gridded ruminant density ($D_{grazing,m,k}$) has been re-calculated based on modified $D_{m,k}$, while the description of calculation is the same as that in the previous manuscript.

**[Comment 7]** (7) This point may be trivial, so it is just a suggestion. I don't think the variable of ME index (Im,j, page 8 and page SI_3) is really necessary unless the ME index has some other meanings. The assumptions seemed just to be: see the supplement. The ME index made the conversions more complicated than it should be.

**[Response]** Thanks for the suggestion. Yes, the ME index ($I_{m,j}$) is not necessary, and might complicate the conversions. Thus we have deleted it in the revised manuscript.

**Appendix A1: Calculation of $Y_{grazed}$ and $Y_{mown}$ in ORCHIDEE-GM v3.1**

The effective yield on grazed grassland ($Y_{grazed}$, unit: kg DM m$^{-2}$ yr$^{-1}$ from grazed grassland) depends on the grazing stocking rate (here, $D_{grazing}$) and on the environmental conditions of the grid cell (Chang et al., 2015a), and calculated as:

$$Y_{grazed,m,k} = IC \times T_{grazing,m,k} \times D_{grazing,m,k}$$

where IC is the daily intake capacity for 1 LU ($\sim$ 18 kg dry matter per day calculated in Supporting information Text S1 of Chang et al., 2015b), $T_{grazing,m,k}$ is the number of grazing days in grid cell k at year m. Due to the impact of livestock on grass growth through trampling, defoliation (i.e., biomass intake) etc., and because grassland cannot be continuously grazed during the vegetation period, thresholds of shoot biomass are set for starting, stopping and resuming grazing (Vuichard et al., 2007). The 'recovery' time required under grazing is obtained in the model using threshold (Vuichard et al., 2007; Chang et al., 2015a), which determine when grazing stops (dry biomass remaining lower than 300 kg DM ha$^{-1}$), or when grazing can start again (dry biomass recovered to a value above 300 kg DM ha$^{-1}$ for at least 15 days. Under mowing, the frequency and magnitude of forage harvests in each grid cell is a function of grown biomass (Vuichard et al., 2007). $Y_{mown,m,k}$ (unit: kg DM m-2 yr-1 from mown grassland) is the annual total harvested grass biomass.

**Appendix A2: Calculation of the historical changes of ruminant stocking density (D)**

Domestic ruminant numbers, and therefore stocking density, are continually changing from year-to-year as reported in FAOSTAT (2014). However, GLW v2.0 only provides livestock density for the reference year (i.e., 2006). To establish the historic changes of ruminant density from 1901 to 2012, two assumptions were made: 1) the distribution of ruminant density did not change during the time-span of this study (1901 - 2012); and 2) the changes in the ruminant density of each category in grid-cell k in country j ($D_{m,j,k}$) co-varied with the changes in category-specific ME requirement in that country. Thus the total ruminant density for grid-cell k in country j in year m ($D_{m,j,k}$) is calculated as:

$$D_{m,j,k} = \sum (D_{ref,i,j,k} \times \frac{ME_{m,i,j}}{ME_{ref,i,j}})$$

where $D_{ref,i,j,k}$ is the ruminant density of category i for grid-cell k in country j in reference year (i.e., 2006); $ME_{m,i,j}$ and $ME_{ref,i,j}$ are the total ME requirement by ruminant category i for country j in year m and in the reference year 2006 respectively. The method to calculate ME requirement is given in Supporting Information Text S1 of Chang et al., 2015b. Here, the range of year m is from 1961 to 2012, since FAOSTAT (2014) provides annual country-averaged statistical data for dairy cows, beef cattle, sheep and goats of livestock numbers (with the unit in head), and meat (carcass weight) or milk yield for the period from 1961 up to the present day.

For the period 1900-1960, regional livestock numbers by 10-year interval derived from Mitchell (1993, 1998a,b) were scaled in 1961 to match the FAOSTAT data (data processed by Dr. Kees Klein

Goldewijk, and given for 17 world regions with the numbers of cattle, sheep and goats; available in the HYDE database: http://themasites.pbl.nl/tridion/en/themasites/hyde/landusedata/livestock/index-2.html). The 17 world regions were designated for global change research, as defined by Kreileman et al. (1998). Linear interpolation is applied to calculate the regional livestock numbers of each year. Assuming the meat (carcass weight) and milk yield for the period of 1900-1960 are the same as that for 1961 from FAOSTAT (2014), total ruminant density for grid-cell $k$ in region $q$ in year $m$ ($D_{m,p,k}$) is then simply extended to 1900-1960 through:

$$D_{m,q,k} = \sum \left( D_{ref,i,q,k} \times \frac{ME_{m,i,q}}{ME_{ref,i,q}} \right)$$

where $D_{ref,i,q,k}$ is the ruminant density of category i for grid-cell k in region $q$ in reference year (i.e., 2006); $ME_{m,i,q}$ and $ME_{ref,i,q}$ are the total ME requirement by ruminant category i for region $q$ in year m and in the reference year 2006 respectively.

**Appendix A3: Calculation of the historical changes of grass biomass use (*GBU*)**

[revised manuscript text omitted]

---

## Author Response (AR1)

The manuscript estimates globally the historical management intensity of grasslands. Thereby, authors use the process-based vegetation model ORCHIDEE-GM in combination with globally derived maps on livestock density, wild herbivory density, nitrogen fertilization and atmospheric nitrogen deposition, and grass-biomass use. Authors can show that largest fractions of managed grasslands occur in regions of high livestock density. A comparison of grassland productivity between managed and unmanaged grassland simulations shows that management has largest impact in regions of high N fertilizer applications. Authors further examined a global increase of 116% of managed grassland area (from $5.1 \times 10^6$ km2 in 1901 to $11 \times 10^6$ km2 in 2000). The topic is interesting and scientifically relevant as more research focusses on the global impact of land use but historical data on land use is rare. Nevertheless, the manuscript requires large improvements.

**[Response]** We thank the reviewer for the valuable comments. Please find our detailed responses below each comment in blue and the corresponding major modifications in the revised manuscript following the response.

**[Comment 1]** I miss a clear statement on the hypothesis or goal of this study in the introduction. While reading the manuscript, it was confusing if authors focus on global management intensity, net biome productivity (NBP) or grassland productivity (NPP). Previous studies and intentions of the study presented in this manuscript are mixed so that it is confusing which parts of this study are novel and which parts are used from previous studies. Is the presented study just an extension of the Europe-study of Chang et al. 2015a? Which challenges arise by constructing a management intensity map for the globe instead of only Europe? Are there differences in the methodology? I highly recommend (1.) providing a clear statement on the goal of this study, (2.) highlighting challenges which arise and (3.) indicating the authors' own novel contribution for achieving this goal. The results and discussion section should also be more focused, following the hypotheses or goals that should be formulated clearly in the introduction.

**[Response]** Thanks for the suggestion. We have rephrased the 'Introduction' section. In the revised introduction, we presented the importance of grassland management intensity history (paragraph 1 and 2), pointed out the limitations of the previous studies related to grassland management and the lack of the gridded management intensity history maps (paragraph 2). Then we cited a recent study that provides a starting point to the reconstruction in this study (paragraph 3). In the last paragraph of introduction, we presented the goal, and the structure of this study.

This study is beyond an extension of the Europe-study. We pointed out the limitation of previous study (paragraph 2) to emphasize the necessary of gridded information on management intensity and the long-term history (1901-2012), which does not exist before and is the challenge and novelty of this study.

In the revised manuscript, we reorganized the structure to better focus on the major goal of this study as reconstructing the history of grassland management intensity. Given the fact that the gridded grassland management intensity maps are productivity-dependent, we still give a specific attention to the evaluation of modelled productivity against both a new set of site-level NPP measurements, and satellite-based models of NPP and GPP. The evaluation part has been combined and shortened in the revised manuscript.

**[Comment 2]** Besides the motivation of this study, the methods section requires large clarification in a similar way. For the model description the authors write about applications of recent model versions (v1 and v2.1) and state that they use version 3.1 of ORCHIDEE-GM. However, I would expect (especially for readers who are not familiar with ORCHIDEE and ORCHIDEE-GM) to get basic information on the model (i.e. most important modelled processes, time step, spatial scale, important input and output of the model).

**[Response]** In Sect. 2.1 of the revised manuscript, we have added some more basic information of ORCHIDEE-GM including the processes and output of the management module and the time step. The spatial scale is presented in the previous manuscript as "from site-level to global scale". ORCHIDEE is able to simulate "carbon fluxes, and water and energy fluxes from site-level to global scale", and the detail processes can be found in the model description paper (Krinner et al., 2005). The important input of the model in this study was presented in the Sect. 2.3 'Model input' of the revised manuscript.

**[Comment 3]** Concerning the model parameters in section 2.2, only 2 parameters are mentioned. Information on where to find the other parameters of the model and their values should be provided. Moreover, this paragraph occurs a second time in the supplement (which is just redundant information). The text S1 in the supplement is, however, written much better and more concise than in the main manuscript.

**[Response]** Original Sect. 2.2 in the previous manuscript and Text S1 has been combined as the Sect. 2.2 in the revised manuscript. In addition, the reference on the other model parameterization was added as "All other parameters of ORCHIDEE model are kept consistent with that in Trunk.rev2425. The parameter settings for grassland management module are in consistent with that in ORCHIDEE-GM v1 (Chang et al., 2013) and v2.1 (Chang et al., 2015ab)".

**[Comment 4]** This applies also for the other text paragraphs in the manuscript of section 2.3 and their corresponding text in the supplement. Partly, introductory information occurs in the supplementary paragraph while it is needed in the paragraph of the main manuscript. In turn, technical information occurs in the main manuscript which is hard to understand without reading the supplementary text first.

**[Response]** Original Sect. 2.3 in the previous manuscript has been separated to 2 sections: 2.3 Model input; and 2.4 Simulation set-up. The paragraphs have been rephrased with introductory information and only necessary technical information, and the corresponding text in the supplementary information is reorganized and rephrased too.

**[Comment 5]** Following sections 2.4 and 2.5, it's difficult to understand which maps provide input for ORCHIDEE-GM simulations and which maps are combined with simulation output of ORCHIDEE-

GM. In total, the entire methods section needs large improvements, i.e. clear, concise and comprehensive statements in order to be able to reproduce the results of this study.

[Response] In the revised manuscript, we have added a new flowchart (Fig. 1) illustrating the procedures for reconstructing the management intensity history, and a table (Table 1) listing all variables shown in the method section (including abbreviation, units, related equations, and data sources). We believe that the flowchart and the revised Sect. 2.3-2.5 presented the reconstruction of the grassland management intensity maps in a more comprehensive way than before.

[Comment 6] Regarding the manuscript language and style, I highly recommend to shorten the manuscript and to be more concise and precise, but still comprehensive. The entire manuscript is too long. Sentences are too long to fluently read the manuscript, some paragraphs are too technical. There are grammar and spelling mistakes. References should be double-checked (e.g., page 4, line 12).

[Response] In the revised manuscript, we have reorganized the manuscript through 1) rephrasing Sect. 2.3 'Model input' with only introductory information and necessary technical information; 2) combining the previous Sect. 2.5 'Modelled productivity', 2.6 'Datasets for model evaluation' and Sect. 2.7 'Model-data agreement matrics' as Sect. 2.6 'Model evaluation' in the revised manuscript, 3) combining the model evaluation sections (Sect. 3.2, 3.4 – 3.6 in the previous manuscript), and 4) shortening the discussion on productivity evaluation (Sect. 4.3). The size of this manuscript is reduced from 9656 words to 8713 words (by 9.8%; including abstract, main text and acknowledgement). Furthermore, we paid more attention on the reconstruction part and less on the model evaluation. The number of figures of the revised manuscript has been reduced from 13 to 10.

Thanks for the suggestions. We have corrected the grammar and spelling mistake and double-checked the reference in the revised manuscript. For example, the reference for PaSim model has been corrected as Riedo et al., 1998; Vuichard et al., 2007a,b; Graux et al., 2011. We have shortened or separated some long sentences to present them more clearly.

[Comment 7] The last sentences of the abstract (page 2, lines 13-21) are confusingly written and hard to understand without reading the entire article.

[Response] Given the reason that "the gridded grassland management intensity maps are model-dependent because they depend on modelled productivity", we gave a specific attention to the evaluation of modelled productivity in this study. We have deleted some detail information, and rephrased the last sentences of the abstract.
General comments: This study attempted to reconstruct the history of grassland management by integrating grazing-ruminant stocking density maps, wild-herbivores population density maps, nitrogen fertilizer application maps as well as nitrogen deposition maps to develop grassland management intensity maps. This land use information is very important to global change studies and very interesting as well. The attempt of integrating those scattered data in various scales is valuable even though the methods might be over-simplified. The manuscript, however, poorly delivered this information. I think the title of this manuscript delivered interesting and clear information about the study, but the main text lost focus that were specified in the title and the abstract. The method sections (in both the main text and the SI) are very confusing and could be more organized. Some descriptions on modeling or calculation were unnecessarily complicated, and some assumptions for extrapolating data need to be checked carefully. Overall, the current version requires major revisions before considered for publication.

**[Response]** We thank the reviewer for the valuable comments. Please find our detailed responses below each comment in blue and the corresponding major modifications in the revised manuscript following the response.

**[Comment 1]** (1) 'Results' and 'Discussion' of the current version made this manuscript read like evaluating the performance of the updated version of ORCHIDEE-GM model that includes livestock data to estimate global grass biomass. The model is a key piece in this study, which generates the NPP and GPP, but it seems the goal of this study is actually 'combining livestock production' and 'to reconstruct the history of grassland management'. If so, the main text should be reorganized. The evaluation-related sections could be combined.

**[Response]** Thanks for the suggestions. In order to stick to the goal of this study, we have revised the manuscript through 1) reorganizing Sect. 2.3 - 2.5 to present the procedures of reconstructing grassland management intensity maps more clearly; 2) combining the method sections on model evaluation to one section (2.6); 3) combining and shortening the result sections on model evaluation (Sect. 3.2, 3.4 – 3.6 in the previous manuscript); and 4) shortening the discussion on model evaluation (Sect. 4.3).

**[Comment 2]** (2) The model-related descriptions in the 'Material and Methods' section are not clear. At page 4 line 28-32, it is not clear what was updated in the model v3.1. Only bug-corrections? Are there any updates in modeling ecological processes or management activities?

**[Response]** In the version 3.1 of ORCHIDEE-GM, we made the adjustment of its parameters for the C4 grassland biome (Sect. 2.2), and implemented a specific strategy for wild animal grazing (Sect. 2.3). Furthermore, in the revised manuscript, version 3.1 has been updated with ORCHIDEE Trunk.rev2425 (a recent version of ORCHIDEE). The above information has been added in the Sect. 2.1 of the revised manuscript.

**[Comment 3]** (3) At page 5 line 22-25, the author listed the input data, but the output was never clearly described in the manuscript. This information may be described in previous publications, but it would be good to briefly describe in this manuscript. Line 12-15 at page 7 reads like descriptions of output, but confusing. I think this part is very important as it is related to how the authors defined and quantified 'management intensity', so it needs to be clearly presented.

**[Response]** We have reorganized the sections in the revised manuscript to clarify the model input (Sect. 2.3), simulation set-up (Sect. 2.4), and the procedures for reconstructing management intensity history (Sect. 2.5). Moreover, we have added a new flowchart (Fig. 1) illustrating the procedures for reconstructing management intensity history, and a table listing all variables shown in the method section (including abbreviation, units, related equations, and data sources). We believe that the flowchart and the revised Sect. 2.3-2.5 presented the reconstruction of the grassland management intensity maps in a more comprehensive way than before.

**[Comment 4]** (4) Does '. . . not use a land-cover map in the simulations, but rather consider that grasslands are distributed all over the world' mean the areas that are not characterized as grassland in a land-cover map have zero grass productivities in your productivity maps?

**[Response]** During post-processing, the grids with zero grassland in the land-cover maps ($A_{grass,m,k} = 0$) will be masked, thus will have zero grass productivities in the productivity maps as shown in Fig. 2 in the previous manuscript. This clarification has been added in the Sect. 2.4 'Simulation set-up' of the revised manuscript.

**[Comment 5]** (5) Line 14-15 at page 8, how the Ygrazed is calculated from Dgrazing,m,k? I think this is a key step of this study and should be described clearly. [Variables, equations and data conversions] There are many equations and data conversions in this manuscript. The authors should define variables clearly and present units for important variables (e.g. D in text S2), so that the readers can easily follow the ideas of producing those data sets. Or, a table listing those variables and associated data sources might be helpful.

**[Response]** Thanks for your suggestion. We added the description about how the model calculates the $Y_{grazed}$ and $Y_{mown}$ in the revised manuscript to clarify this key step. We also added a new table listing all variables shown in method section, including abbreviation, units, related equations, and data sources (Table 1).

**[Comment 6]** (6) I think the assumption at Line 4-5 at page SI_3 might be wrong as the ratio of the total ruminant density between years can be calculated based on the assumptions in text S2. I could be wrong, but I think the authors should carefully check the conversion and should not make too many assumptions arbitrarily as this might affect the results significantly. A brief interpretation of my thoughts: see the supplement for equations and calculations.

**[Response]** Thank you for the comment. Yes, you are right about the calculation. We should calculate the gridded ruminant density ($D_{m,k}$) variation and gridded grass biomass use ($GBU_{m,k}$) based on the category-specific variation of metabolisable energy (ME) requirement in the country rather than the changes in country-scale total ME requirement. Thus we have modified all related calculations

(including $D_{m,k}$, $D_{grazing,m,k}$, and $GBU_{m,k}$), re-run all simulations, and re-calculate gridded management intensity history based on modified calculation. In the revised manuscript, the calculations of $D_{m,k}$ and $GBU_{m,k}$ have been changed accordingly. The gridded ruminant density ($D_{grazing,m,k}$) has been re-calculated based on modified $D_{m,k}$, while the description of calculation is the same as that in the previous manuscript.

**[Comment 7]** (7) This point may be trivial, so it is just a suggestion. I don't think the variable of ME index (Im,j, page 8 and page SI_3) is really necessary unless the ME index has some other meanings. The assumptions seemed just to be: see the supplement. The ME index made the conversions more complicated than it should be.

**[Response]** Thanks for the suggestion. Yes, the ME index ($I_{m,j}$) is not necessary, and might complicate the conversions. Thus we have deleted it in the revised manuscript.

**List of major changes in the revised manuscript**

1. Sect. 1 'Introduction' has been reorganized and rephrased to clearly present the goal and the novelty of this study.

2. Original Sect. 2.2 in the previous manuscript and Text S1 has been combined as the Sect. 2.2 in the revised manuscript to precisely present the model parameter calibration.

3. Original Sect. 2.3 in the previous manuscript has been separated to 2 Sect.s: 2.3 Model input; and 2.4 Simulation set-up. The paragraphs have been rephrased with introductory information and only necessary technical information, and the corresponding text in the supplementary information is reorganized and rephrased too.

4. In Sect. 2.5, we have added a new flowchart (Fig. 1) illustrating the procedures for reconstructing the management intensity history, and a table (Table 1) listing all variables shown in the method section (including abbreviation, units, related equations, and data sources). We believe that the flowchart and the revised Sect. 2.5 presented the reconstruction of the grassland management intensity maps in a more comprehensive way than before.

5. The revised Sect. 2.6 "Model evaluation: datasets and model-data agreement metrics" is the combination of previous Sect. 2.5 – 2.7 with only necessary information.

6. The site-level NPP dataset has been updated with 16 sites across western Siberia. The data providers have been added as new co-authors given their contribution on evaluation data and the valuable comments on the revised manuscript.

7. Due to a corrected calculation of input maps (including $D_{m,k}$, $D_{grazing,m,k}$, and $GBU_{m,k}$), new simulations were carried out resulting in new output in Sect. 3.

8. The result sections on model evaluation (Sect. 3.2, 3.4 – 3.6 in the previous manuscript) have been combined as Sect. 3.3, and shortened with concise expressions. Several nonessential results and corresponding discussions have been deleted, such as the magnitude of the GPP IAV (coefficient of variation, CV) and the maximum monthly GPP ($GPP_{max}$).

[revised manuscript text omitted]

* * *
Jinfeng Chang 30/4/2016 12:57

Jinfeng Chang 30/4/2016 16:39

Jinfeng Chang 29/4/2016 11:19

Jinfeng Chang 29/4/2016 11:31

Jinfeng Chang 17/5/2016 10:09
**Moved (insertion) [2]**

Jinfeng Chang 17/5/2016 10:06

Jinfeng Chang 17/5/2016 10:07

Jinfeng Chang 17/5/2016 10:09
**Moved up [2]:** Chang et al. (2015b) then added a parameterization of adaptive management through which farmers react to a climate-driven change of previous-year productivity. Though a full nitrogen cycle is not included in ORCHIDEE-GM, the positive effect of nitrogen fertilizers on grass photosynthesis rates, and thus on subsequent ecosystem productivity and carbon storage, are parameterized with an empirical function calibrated from literature estimates (Chang et al., 2015b).

Jinfeng Chang 17/5/2016 10:15

ORCHIDEE-GM was applied to simulate GHG budgets and ecosystem carbon stocks under climate, $CO_2$ and management changes for Europe. But an extension of model application to regions outside Europe requires first a calibration of key productivity related parameters. Two sensitive parameters representing photosynthetic capacity (the maximum rate of Rubisco carboxylase activity at a reference temperature of 25°C; $Vc_{max}25$) and the morphological plant traits (the maximum specific leaf area; $SLA_{max}$) were reported by Chang et al. (2015a) for simulating grassland NPP. The $Vc_{max}25 = 55$ µmol $m^{-2}$ $s^{-1}$ and $SLA_{max} = 0.048$ $m^2$ per g C in ORCHIDEE-GM were previously defined from observations and indirectly evaluated against eddy-flux tower measurements of GPP for temperate C3 grasslands in Europe (Chang et al., 2013, 2015b). The global TRY database gives SLA values for C4 grasses, of 0.0192 $m^2$ $g^{-1}$ dry matter (0.0403 $m^2$ per g C with a mean leaf carbon content per dry matter of 47.61%; Kattge et al., 2011). Thus, we have set the value of $SLA_{max} = 0.044$ $m^2$ per g C for C4 grasses in ORCHIDEE-GM to fit the mean value from the TRY estimate, as we did previously for C3 grasses (Chang et al., 2013). The parameter $Vc_{max}25$ cannot be directly measured, but it is usually derived from $A/C_i$ curves in C3 or C4 photosynthesis models (C3: Farquhar et al., 1980; C4: Collatz et al., 1992) where A is the leaf-scale net $CO_2$ assimilation rate and $C_i$ the partial pressure of $CO_2$ in leaf intercellular spaces. Several researches provide observation-based estimates of $Vc_{max}25$ (Feng and Dietze, 2013; Verheijen et al., 2013; range of 24 – 131 µmol $m^{-2}$ $s^{-1}$ for C3 grasses, and of 15 – 46 µmol $m^{-2}$ $s^{-1}$ for C4 grasses). Based on these estimates, we keep the value of $Vc_{max}25 = 55$ µmol $m^{-2}$ $s^{-1}$ previously calibrated in Europe for all C3 grasses, and set $Vc_{max}25 = 25$ µmol $m^{-2}$ $s^{-1}$ for C4 grasses. These values may not reflect differences in nitrogen, and phosphorus availability between locations, nor adaptation or species changes within a C3 or C4 grassland, but they are within the range of observations made under different conditions, and consistent with values used by other terrestrial ecosystem models (Table S1). All other parameters of ORCHIDEE model are kept the same as in Trunk.rev2425. The parameter settings for grassland management module are in consistent with that in ORCHIDEE-GM v1 (Chang et al., 2013) and v2.1 (Chang et al., 2015a, b).

**2.3 Model input**

[revised manuscript text omitted]

Jinfeng Chang 13/5/2016 11:18